## RESEARCH ARTICLE

# Piezo1 balances membrane and cortex tension to stabilize intercellular junctions and maintain the epithelial barrier

Ahsan Javed[1], Aki Stubb[1,2], Clémentine Villeneuve[1,2], Satu-Marja Myllymäki[1], Franziska Peters[1,2], Matthias Rübsam[3], Carien M. Niessen[3], Leah C. Biggs[1,2] and Sara A. Wickström[1,2,4,*]

## ABSTRACT

Formation of the skin barrier is essential for organismal survival and tissue homeostasis. Barrier formation requires positioning of functional tight junctions (TJs) to the most suprabasal viable layer of the epidermis through a mechanical circuit that is driven by generation of high tension at adherens junctions. However, what allows the sensing of tension build-up at these adhesions and how this tension is balanced to match the requirements of tissue mechanical properties is unclear. Here, we show that the mechanosensitive ion channel Piezo1 is essential for the maturation of intercellular junctions into functional continuous adhesions. Deletion of Piezo1 results in an imbalance of cell contractility and membrane tension, leading to a delay in adhesion maturation. Consequently, the requirement for Piezo1 activity can be bypassed by lowering contractility or elevating membrane tension. *In vivo*, Piezo1 function in adhesion integrity becomes essential only in aged mice where alterations in tissue mechanics lead to impaired tight junctions and barrier dysfunction. Collectively these studies reveal an essential function of Piezo1 in the timely establishment and maintenance of cell–cell junctions within a mechanically tensed epidermis.

KEY WORDS: Piezo1, Epidermis, Cell–cell contacts, Mechanotransduction

## INTRODUCTION

Epithelial cells line the surfaces of organs and generate a barrier to protect underlying tissues and maintain the physiological organ environment. Consequently, barrier dysfunction causes diseases ranging from infections to chronic inflammation and understanding the central mechanisms of barrier function is crucial for the development of treatments to restore or maintain barriers (Arwert et al., 2012; Kubo et al., 2012). The skin epidermis is a multilayered stratified tissue that serves as the primary barrier between the body and the external environment. In mammalian skin, the epidermis has two sets of physical barriers, the stratum corneum and tight junctions (TJs). The stratum corneum is the outermost epidermal layer, mainly consisting of terminally differentiated corneocytes and intercellular lipids. TJs are intercellular junctions present in the uppermost viable layer stratum granulosum II (SG2) (Rübsam et al., 2018). Together, these two barriers protect the body from environmental stresses such as irradiation, pathogens, chemical irritants and mechanical stress, while also preventing water loss (Blanpain and Fuchs, 2006, 2009; Candi et al., 2005; Koster and Roop, 2007).

The formation of intercellular junctions is initiated by formation of so-called primordial adherens junctions that form an interdigitated 'adhesion zipper' at the cell–cell interface. These zipper-like adhesions then evolve into spatially separated, continuous 'belt-like' adherens junctions and TJs coinciding with epithelial polarization (Vasioukhin et al., 2000; Yano et al., 2017). TJs consist of claudin adhesion receptors, which polymerize into a network of intercellular strands creating the diffusion barrier (Tsukita et al., 2001). On their cytoplasmic side, the zona occludens proteins (ZO) 1 and 2 (also known as TJP1 and TJP2, respectively), which initially are already present in primordial adherens junctions, form a membrane-attached scaffold to facilitate formation and sub-apical positioning of claudin strands. In simple epithelia, claudin receptor oligomerization enables condensation of the scaffold protein ZO-1 that then, likely through recruitment of other cytoskeletal and signaling proteins, promotes local actin polymerization and bundling to form a TJ belt (Fanning and Anderson, 2009; Pombo-García et al., 2024). The tethering of claudins to actin through ZO-1 is dynamic and transient, likely facilitating cell shape adaptations during motility (Citi, 2019; Van Itallie et al., 2009). Importantly, the molecular mechanisms underlying the spatial assembly and formation of TJs from primordial adherens junctions involves an E-cadherin-driven mechanical signaling network that regulates the organization of the actomyosin cortex to stabilize TJs. Depletion of cadherins leads to decreased cortical tension and the inability of cells to form TJs (Brückner et al., 2019; Rübsam et al., 2017). However, how cortical tension and contractility is sensed and balanced locally at cell–cell junctions is unclear.

Piezo1 is a mechanosensitive ion-channel located primarily at the plasma membrane and has been demonstrated to be activated by several mechanical stimuli such as stretching, shear stress or by sensing changes in local curvature of the plasma membrane (Albarrán-Juárez et al., 2018; Li et al., 2014; Maneshi et al., 2018; Ranade et al., 2014; Rode et al., 2017; Wang et al., 2016; Wu et al., 2017). Piezo1 has been shown to play a prominent role as a mechanosensor in development, homeostasis and disease (Murthy et al., 2017). In the epithelium of *Xenopus* Piezo1-mediated $Ca^{2+}$ flashes have been implicated in maintaining an intact TJ barrier by

[1]Stem Cells and Metabolism Research Program, Faculty of Medicine, University of Helsinki, 00290 Helsinki Finland. [2]Department of Cell and Tissue Dynamics, Max Planck Institute for Molecular Biomedicine, 48149 Münster, Germany. [3]Department Cell Biology of the Skin, Cologne Excellence Cluster on Cellular Stress Responses in Aging Associated Diseases (CECAD), Center for Molecular Medicine Cologne, University Hospital Cologne, University of Cologne, Joseph-Stelzmann-Str. 26, 50931 Cologne, Germany. [4]Helsinki Institute of Life Science, Biomedicum Helsinki, University of Helsinki, 00290 Helsinki, Finland.

*Author for correspondence (sara.wickstrom@mpi-muenster.mpg.de)

M.R., 0000-0002-0012-2601; L.C.B., 0000-0002-4990-8664; S.A.W., 0000-0001-6383-6292

regulating cortical actomyosin contraction (Varadarajan et al., 2022). At the molecular level, Piezo1 has shown to localize at cell–matrix adhesions (Yao et al., 2022) and adherens junctions, where it has demonstrated to interact with E-cadherin, β-catenin and filamentous (F)-actin to regulate barrier function and permeability (Chuntharpursat-Bon et al., 2023; Jiang et al., 2021; Wang et al., 2022). The role of Piezo1 in cell–cell junctions of the mammalian skin epidermis has not been studied.

Here, we address the mechanisms by which cortical and membrane tension are balanced to facilitate formation and mechanical stability of intercellular junctions and the role of Piezo1 in this process. We show that timely transition of zipper-like primordial junctions into a mature, continuous, occludin-containing TJs requires Piezo1 mechanosensing. In the absence of Piezo1, intercellular junctions exhibit delayed maturation and decreased mechanical stability, which can be bypassed by modulating membrane or cortical tension. Consequently, *in vivo*, in the context of aging epidermis that exhibits elevated tissue stiffness, the absence of Piezo1 leads to the impairment of epidermal barrier function. Collectively these data demonstrate a role for Piezo1 in balancing membrane tension and cortical tension, which is ultimately required to stabilize intercellular junctions and TJs to maintain skin barrier integrity.

## RESULTS

### Intercellular junction maturation requires balancing cortex and membrane tension

To investigate the morphological changes at the cell–cell adhesion interface during intercellular adhesion maturation, we performed live imaging of mouse primary epidermal keratinocytes after switching cells to high $Ca^{2+}$ (Fig. 1A; Movie 1). Using ZO-1–mEmerald as a marker, we quantified the transition from zipper-like primordial intercellular junctions into continuous 'belt-like' junctions. We observed a temporally heterogeneous process with initiation of belt formation 80 min after adding $Ca^{2+}$ into the growth medium, that was completed in all cells by 16 h post $Ca^{2+}$ switch, concomitant with abrupt cellular flattening upon transition from zippers to belt-like adhesions (Fig. 1A; Movie 1).

Cell shape is mainly controlled by two opposing forces – cortical tension, which reduces cell contacts, and intercellular adhesion forces, which increase the surface of contacts (Lecuit and Lenne, 2007). We thus sought to analyze how these forces are balanced in space and time during adhesion maturation in keratinocytes. Intercellular junctions can recruit cortical actin filaments directly, via α-catenin, or indirectly through proteins such as vinculin (Budnar and Yap, 2013; Fierro-González et al., 2013; Maître et al., 2012; Rübsam et al., 2017; Wu and Yap, 2013). Myosin activity and specifically non-muscle myosin IIA (NMIIA) then generates the mechanical tugging force necessary for cell–cell junction reinforcement and maintenance (Heuzé et al., 2019). At 4 h post $Ca^{2+}$ application, apical F-actin was found organized mainly in stress fibers and punctate junctional actin associated with zippers (Fig. 1B,C). Concomitant with emergence of continuous junctions (8 h), the stress fibers were replaced by thick actin bundles positioned parallel to junctions (Fig. 1B,C). In contrast to stress fibers, the actin at junctions was low in myosin activity (phosphorylated myosin light chain; pMLC2, encoded by *MYL9*) (Fig. 1B,C).

We next approximated tension within the junctions themselves by quantifying their molecular links to actin – vinculin and α-catenin (Fig. 1D). Quantification of junctional vinculin intensity at the early

zipper-like stage and mature belt-like stage indicated an increase in vinculin abundance (Fig. 1D). Additional analysis of the tension-sensitive epitope (α18)-catenin (Yonemura et al., 2010) during the time course of maturation process confirmed the build-up of junctional stresses concomitant with adhesion maturation (Fig. S1A). This indicated that formation of belt-like intercellular adhesions was associated with initial contractility build-up by actomyosin stress fibers linked to junctions, followed by a switch to parallel actomyosin bundles and reduced contractility at adhesions, while the junctions themselves were stabilized in a tensed state, as indicated by vinculin recruitment as a proxy for a strengthened actin-junction link.

The results so far are consistent with previous observations that cell contractility, and thus cortical tension, reduces the length of cell junctions, whereas cell–cell adhesion tends to extend the length of the junctions (Lenne et al., 2021; Maître and Heisenberg, 2013; Manning et al., 2010). Although these models focus on cytoskeleton-based mechanisms, effective surface tension is also controlled by membrane tension (Sitarska and Diz-Muñoz, 2020). Consistent with this, increased membrane tension has been shown to promote contacts between neighboring cell membranes to allow cadherin clustering (Delanoë-Ayari et al., 2004). Actomyosin-mediated cortex tension has further been shown to anticorrelate with membrane tension (Gauthier et al., 2011; Houk et al., 2012). We thus addressed changes in plasma membrane tension during junction maturation. To this end we performed fluorescence-lifetime imaging (FLIM) of the membrane tension reporter FLIPPER-TR (Colom et al., 2018), accompanied by controls measuring lifetimes after hyper and hypotonic stress (Fig. S1B). FLIM imaging revealed an increase in FLIPPER-TR lifetime during the TJ maturation phase, indicative of increasing membrane tension (Fig. 1E).

To test whether balancing actomyosin and membrane tension are coordinating timely formation of continuous cell–cell junctions, we first inhibited myosin activity using blebbistatin (Rheinlaender et al., 2020) or artificially elevated membrane tension by manipulating osmolarity during the time course of junction formation (Charras et al., 2005). Indeed, blebbistatin treatment (added 6 h before fixation) accelerated the transition of junctions from the immature zipper-like state to a continuous mature state (Fig. 1F). Consistent with this, elevating membrane tension by inducing cell swelling with a mild hypotonic treatment, added 30 min before fixation, led to accelerated maturation of cell–cell junctions (Fig. 1G). Collectively, these data showed that actomyosin cortex tension and membrane tension play a crucial role in coordinating the dynamics of cell–cell junctions transitioning from a zipper-like primordial state into a mature belt-like configuration.

### Piezo1 regulates maturation dynamics of cell–cell junctions

To address the molecular mechanisms by which contractility and membrane tension are regulated upon intercellular junction maturation, we hypothesized that this would require their local sensing at the junction. A strong molecular candidate is the tension-sensitive ion channel Piezo1. To test the involvement of Piezo1, we generated an epidermis-specific Piezo1 knockout-mouse by crossing Piezo1 floxed/floxed mice (Cahalan et al., 2015) with a keratin14-promoter-driven Cre line (Hafner et al., 2004) (from here on termed Piezo1-eKO). Primary epidermal keratinocytes from Piezo1-eKO and control mice were transfected with ZO-1–mEmerald and live imaged. Quantification of cell–cell junction formation upon $Ca^{2+}$ switch showed significantly delayed dynamics

Journal of Cell Science

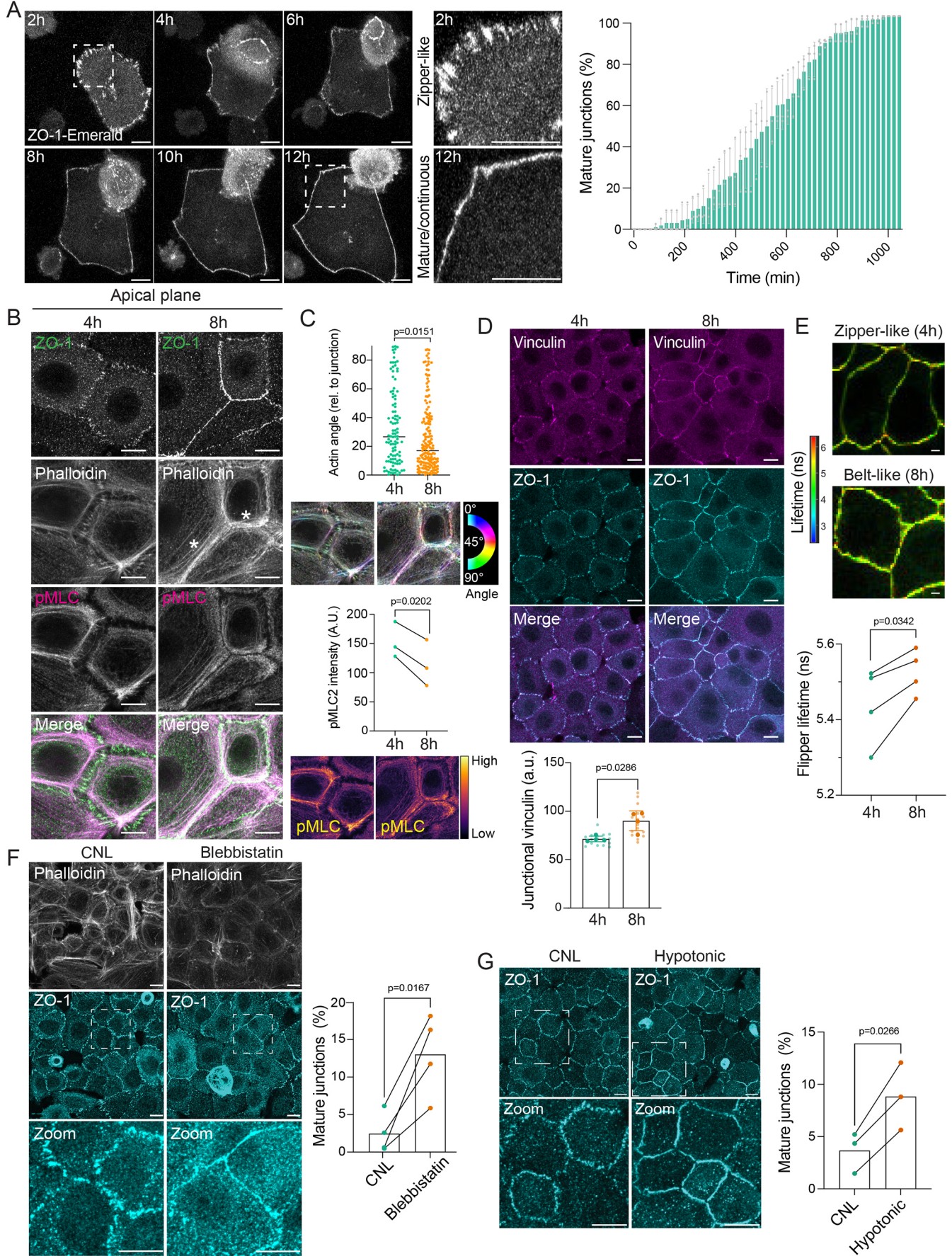

**Fig. 1.** See next page for legend.

**Fig. 1. Intercellular junction maturation requires balancing membrane and cortical tension.** (A) Representative snapshots from live imaging data from ZO-1–mEmerald-expressing keratinocytes post $Ca^{2+}$ switch at time points indicated. Dashed boxes indicate magnified areas (right panels). Quantification shows the proportion of cells with continuous junctions as a function of time ($n$=3 independent experiments with >10 cells per experiment per condition; mean±s.d.). (B,C) Representative images (B) and quantification (C) of apical surface projections of ZO-1, pMLC2 and phalloidin-stained primary keratinocytes fixed at 4 h and 8 h post $Ca^{2+}$ switch. Note redistribution of actin into peripheral bundles oriented in parallel to junctions at 8 h (asterisk). Colors in C indicate orientation angles (upper panel) or intensities (lower panel). C show data for $n$=3 independent experiments with >30 cells per experiment per condition, Kolmogorov–Smirnov (upper panel), paired two-tailed $t$-test (lower panel). (D) Representative images of apical surface projections and quantification of vinculin (and ZO-1-stained primary keratinocytes fixed at 4 h or 8 h after $Ca^{2+}$ addition). Note increase in junctional vinculin intensity at 8 h ($n$=3 independent experiments with >50 cells per experiment per condition; mean ±s.d.; Mann–Whitney). (E) Representative images and quantification of fluorescence lifetime imaging of FLIPPER-TR. Note increase in FLIPPER-TR lifetime indicative of elevated membrane tension after 8 h $Ca^{2+}$ ($n$=4 independent experiments with ≥50 membrane measurements per condition per experiment; paired two-tailed $t$-test). (F) Representative images of ZO-1-stained cells treated with 5 µM blebbistatin. Dashed boxes indicate magnified areas (bottom panels). Quantification shows the faster maturation of junctions in blebbistatin treated cells ($n$=4 independent experiments with >50 cells per experiment per condition; paired $t$-test). (G) Representative images of ZO-1-stained primary keratinocytes exposed to hypotonic medium (140 mOsm). Dashed boxes indicate magnified areas (bottom panels). Quantification shows accelerated maturation of cell–cell junctions in cells exposed to hypotonic medium ($n$=3 independent experiments >100 cells per experiment per condition; paired two-tailed $t$-test). All scale bars: 10 µm. A.U., arbitrary units.

of junctional maturation in Piezo1-eKO cells (Fig. 2A). Although early zipper-like intercellular adhesion formation was comparable between Piezo1-eKO and control cells, Piezo1-eKO cells showed a substantial delay in transitioning into mature belt-like junctions (Fig. 2A; Movie 2). In addition, the Piezo1-eKO cells exhibited more dynamic cytoskeletal motion (Fig. 2A,B; Movie 2 and 3). Analysis of vinculin levels at intercellular junctions showed reduced vinculin intensity at cell–cell contacts, but abundant vinculin at cell–matrix adhesions (Fig. S2A), further indicating lack of aberrant contractility and proper tension build up at intercellular junctions of Piezo1-eKO cells. To more directly quantify tension at the junctions, we performed laser ablation 8 h post $Ca^{2+}$ switch. As predicted by reduced vinculin recruitment, junctions in Piezo1-deficient cells were under lower tension than in control cells (Fig. S2B).

To confirm that this phenotype was not related to expression of exogenous, tagged ZO-1, we performed additional immunofluorescence experiments at 3, 6, 12 and 48 h post $Ca^{2+}$ switch where we stained cells for endogenous ZO-1. Immunofluorescence analyses of fixed cells confirmed delayed maturation of Piezo1-eKO intercellular junctions 12 h post $Ca^{2+}$ switch, and a delay in formation of occludin-positive TJ belts (Fig. S2C,D). Importantly, the delayed maturation phenotype caused by the lack of Piezo1 was rescued by reconstituting Piezo1 expression using Piezo1–FLAG, indicating that Piezo1 controls the maturation process (Fig. 2C), whereas in control cells junction maturation was unaffected by Piezo–FLAG expression (Fig. S2E).

To understand whether Piezo1 exerts its effect on intercellular junctions by directly localizing to these sites, we analyzed the distribution of Piezo1 during the time course of adhesion maturation. In the absence of well-functioning antibodies, we assayed the localization of the Piezo1–FLAG in the rescued

Piezo1-eKO cells. Detection of Piezo1 with anti-FLAG antibody during the time course of cell–cell junction maturation revealed that prior to junction maturation (4 h post $Ca^{2+}$ switch), Piezo1 was enriched at focal adhesion-like structures together with vinculin (Fig. 2D,E,I). Interestingly, Piezo1 also appeared to be partially co-distributed with ZO-1 at zipper-like adhesions at 8 h post $Ca^{2+}$ switch (Fig. 2F,I; Fig. 2E). However, after the junctions had matured (12 h post $Ca^{2+}$ switch), Piezo1 no longer localized to the belt-like junctions but remained associated with E-cadherin-positive adherens junction (Fig. 2G–I). Collectively, these data indicate that Piezo1 is required for timely cell junction maturation into junction belts, and is localized to immature zipper-like cell–cell junctions, adherens junctions and focal adhesions, but no longer to the mature adhesion belts.

## The requirement of Piezo1 for timely cell–cell junction maturation can be bypassed by modulating membrane and cortical tension

Given that junction maturation requires balancing cortical and membrane tension, we next addressed whether Piezo1 regulates either of these parameters in order to promote timely cell–cell adhesion dynamics. FLIM imaging of the FLIPPER-TR membrane tension probe after 3 h of $Ca^{2+}$ switch revealed reduced lifetimes in Piezo1-eKO cells compared to in the control cells, indicative of lower membrane tension in the absence of Piezo1 (Fig. 3A). Furthermore, we approximated cortex tension of Piezo1-eKO keratinocytes using atomic force microscope-mediated force indentation spectroscopy (AFM) with a spherical bead cantilever to measure cortical stiffness (Haase and Pelling, 2015; Miroshnikova et al., 2018). Coinciding with the timescale of zipper-like adhesion formation at 3 h and 6 h post $Ca^{2+}$ switch, which occurred normally also in Piezo1-eKO cells, loss of Piezo1 did not significantly affect the build-up of cortical stiffness at these time points (Fig. 3B). However, at 12 h post $Ca^{2+}$ switch when control cells had formed mature junction belts and cortical stiffness had plateaued, Piezo1-eKO keratinocytes showed a moderate increase in cortical stiffness (Fig. 3B). Consistent with this, analyses of actomyosin contractility revealed that there was still persistence of actin stress fibers and pMLC2 activity at 8 h post $Ca^{2+}$ switch (Fig. 3C,D), suggesting a delay in balancing cortex tension.

To explore whether the observed imbalance of actomyosin and membrane tension could explain the impaired junctional maturation in Piezo1-eKO cells, we next asked whether reducing cortical contractility or enhancing membrane tension impacted the kinetics of junction maturation. Strikingly, reducing cortical contractility by inhibiting myosin activity with blebbistatin rescued the junctional maturation defect (Fig. 3E). To further understand the mechanism by which reducing contractility rescued junction formation, we inhibited ROCK1, the upstream kinase of myosin. Similar to what was seen with blebbistatin, inhibition of ROCK1 activity using Y27362 added 2 h before fixation restored timely adhesion maturation in Piezo1-eKO cells (Fig. S3A).

Importantly, increasing membrane tension by hypotonic shock also facilitated timely maturation of cell–cell junctions in Piezo1-eKO cells (Fig. 3F). This led us to hypothesize that Piezo1 is required to balance membrane and cortex tension during junction maturation. To test this, we first inhibited cortical contractility using the myosin inhibitor blebbistatin and analyzed membrane tension using the FLIPPER probe at 8 h post $Ca^{2+}$, the time point when Piezo1-eKO cells were still showing immature junctions. Indeed, although blebbistatin treatment (added 2 h before analyses) led to a mild decrease in membrane tension in control cells, this treatment

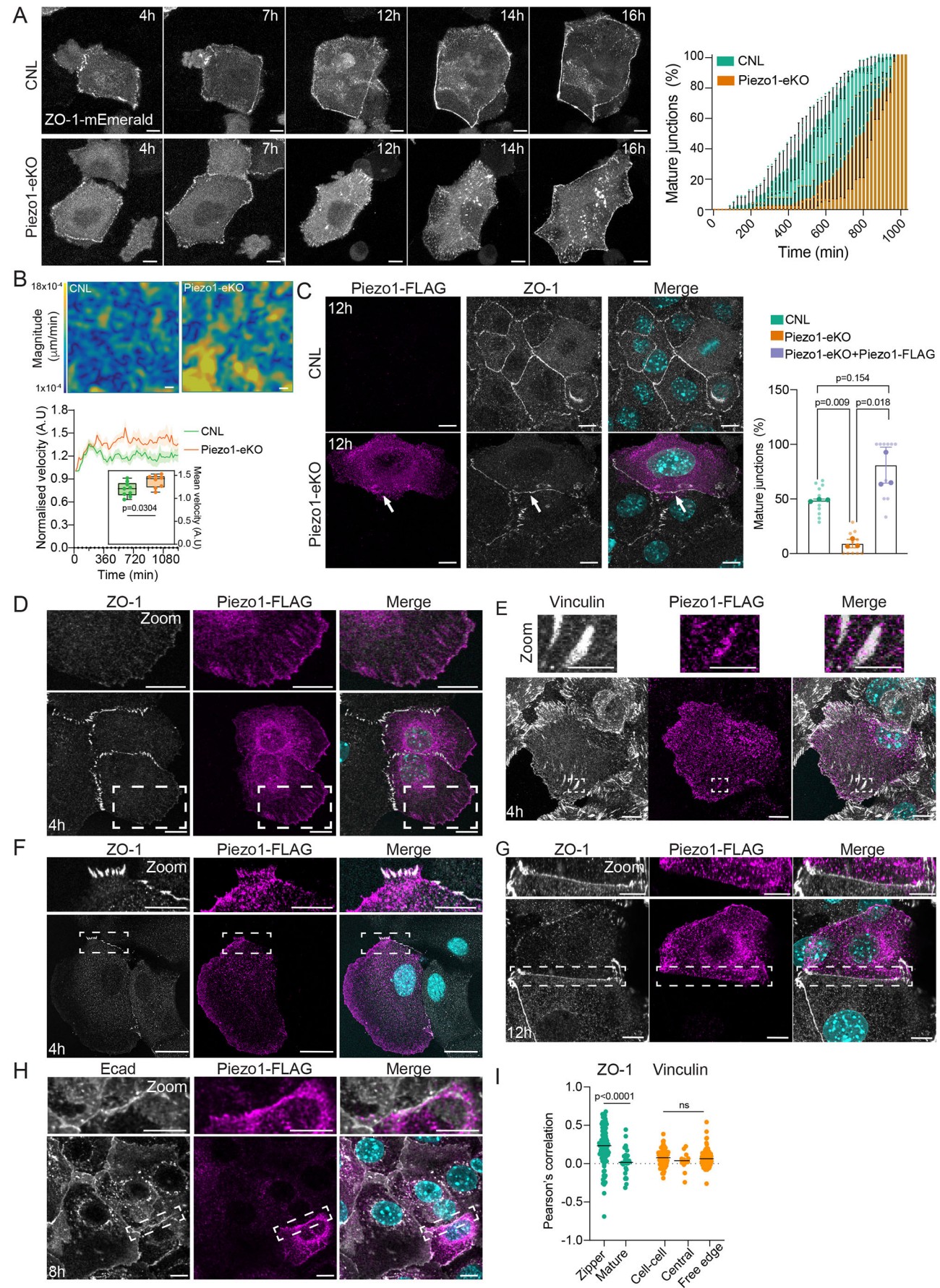

**Fig. 2.** See next page for legend.

**Fig. 2. Piezo1 regulates maturation of cell-cell junctions.**
(A) Representative live imaging snapshots of ZO-1–mEmerald cell–cell junction assembly/maturation in CNL and Piezo1-eKO primary keratinocytes. Quantification shows relative proportion of cells with continuous junctions as a function of time. Note delay in junction maturation of Piezo1-eKO cells ($n$=3 independent experiments with >10 cells per experiment per condition; mean±s.d.). Scale bars: 10 µm. (B) Representative images and quantification of velocity magnitudes and mean velocities of CNL and Piezo1-eKO monolayers quantified by PIV from live imaging of actin-labeled cells ($n$=3 independent experiments with >3 fields of view per experiment per condition; mean±s.e.m. and box plot, where the box represents the 25–75th percentiles, and the median is indicated. The whiskers show minimum-to-maximum range). Scale bars: 20 µm. (C) Representative images of ZO-1- and FLAG-stained CNL and Piezo1-eKO primary keratinocytes transfected with Piezo1–FLAG. Note that expression of Piezo1–FLAG in Piezo1-eKO rescues the dynamics of junction maturation (arrows) ($n$=3 independent experiments with >35 cells per experiment per condition; mean±s.d., one-way ANOVA with Tukey's post test; ns, not significant). Scale bars: 10 µm. (D) Representative images of ZO-1- and FLAG-stained Piezo1-eKO primary keratinocytes transfected with Piezo1–FLAG. Dashed boxes indicate magnified areas (top panels). Note localization of Piezo1–FLAG in focal adhesion-like structures. (E) Representative images of vinculin- and FLAG-stained Piezo1-eKO primary keratinocytes transfected with Piezo1–FLAG. Dashed boxes indicate magnified areas (top panels). (F) Representative images of ZO-1- and FLAG-stained Piezo1-eKO primary keratinocytes transfected with Piezo1–FLAG. Note partial colocalization of ZO-1 and Piezo1–FLAG in zipper-like adhesions. (G) Representative images of ZO-1- and FLAG-stained Piezo1-eKO primary keratinocytes transfected with Piezo1–FLAG. Note lack of colocalization of ZO-1 and Piezo1–FLAG at linear junctions. (H) Representative images of E-cadherin- and FLAG-stained Piezo1-eKO primary keratinocytes transfected with Piezo1-FLAG. Dashed rectangles mark region of zoom-in. All scale bars in D–H: 10 µm (main images); 5 µm (magnifications). Time stamps refer to time post Ca$^{2+}$ switch. Images representative of at least three independent repeats. (I) Quantification of colocalization (Pearson's correlation of pixel intensity) of hPiezo1–FLAG and ZO-1 and vinculin (30 cells pooled across three independent experiments; Kruskal–Wallis and Dunn's test; ns, not significant). A.U., arbitrary units.

triggered an increase in membrane tension in the Piezo1-eKO cells to the level of control cells (Fig. S3B). Vice versa, treating the cells with hypotonic buffer for 30 min prior to AFM measurements of cortex stiffness, showed that increasing membrane tension in Piezo1-eKO cells reduced the cortex stiffness to the level of control cells (Fig. 3G). Collectively these experiments indicate that cortex and membrane tension are inversely correlated, and that reducing contractility in Piezo1-eKO cells normalized the lowered membrane tension and elevating membrane tension reduced cortical stiffness, both facilitating timely junction maturation into belts.

### Aging leads to defects in TJ maturation and barrier function in Piezo1-eKO mice

To understand whether the observed role of Piezo1 in junction dynamics was relevant for epidermal development and homeostasis *in vivo*, we first examined Piezo1 expression within the stratified epidermal layers by combining RNA *in situ* hybridization (RNAscope) with immunofluorescence. Staining of wild-type back skin sections with markers for basal stem cells (keratin-14) and suprabasal differentiated cells (keratin-10) and *Piezo1* RNA revealed Piezo1 expression in both layers, with slightly more abundant *Piezo1* mRNA in the suprabasal layers (Fig. S4A). Consistent with the RNAscope results, re-analyses of published single-cell RNA sequencing data from adult back skin (Joost et al., 2020; GEO: GSE129218) showed that the highest expression of Piezo1 was in the differentiated layers of the epidermis (Fig. S4B).

Furthermore, triggering adhesion maturation and terminal differentiation *in vitro* using a Ca$^{2+}$ switch also led to the increased *Piezo1* mRNA levels (Fig. S4C), indicating that Piezo1 expression is temporally correlated with keratinocyte differentiation and/or junction maturation.

Next, we addressed the role of Piezo1 in regulation of epidermal homeostasis and intercellular junction formation *in vivo* by analyzing Piezo1-eKO skin. Piezo1-deficient skin showed no gross skin or hair phenotype, and histological analysis of back skin from young adult mice (2–6 months old) revealed no morphological differences (Fig. S4D). For optimal visualization of cell-cell junctions, we utilized epidermal whole-mounts (Rübsam et al., 2017). Staining of young adult ear whole mounts revealed very subtle, but not statistically significant differences in TJ morphology as marked by ZO-1 and occludin stainings or in overall cell morphologies (Fig. S4E).

Given the role of Piezo1 in mechanosensing and the increase in tissue stiffness of aged mouse skin (Koester et al., 2021), we proceeded to analyze the changes to the epidermal phenotype during aging. Indeed, at 1 year of age, ear whole-mounts showed reduced junctional ZO-1 intensity within the SG2 layer (Fig. 4A). We also observed reduced levels of vinculin at cell–cell contacts in the SG2 layer of Piezo1-eKO skin (Fig. 4A), similar to the phenotype observed in cell culture. Quantification of SG2 cell shape and TJ morphology revealed more irregular cell shapes as well as fewer straight junctions in aged Piezo1-eKO mice (Fig. 4B,C).

Given the crucial role of TJs in establishing the bi-directional permeability barrier, we proceeded to test the functionality of these abnormal TJs by measuring transepidermal water loss (TEWL) in newborn, young adult and 1-year-old mice. Consistent with the morphologically unaltered TJs, no barrier impairment was observed in the early postnatal stage [postnatal day (P)1–3; Fig. S4F] or young adult mice (Fig. S4G,H). In contrast, and consistent with the abnormal TJs and cell shapes, male and female 1-year-old Piezo1-eKO mice displayed an increase in TEWL, indicating that the barrier function in Piezo1-eKO mice is compromised (Fig. 4D). Consistent with this, levels of claudin-1, a protein essential for epidermal barrier function (Furuse et al., 2002), were mildly and progressively reduced at TJs of young and aged Piezo1-eKO mice (Fig. 4E; Fig. S4I). Collectively, these findings indicate that Piezo1 is required for efficient TJ maturation and barrier function in particular in aged mice, possibly due to increased tissue stiffness.

### Deletion of Piezo1 results in impaired epidermal homeostasis and tissue fragility

To understand whether the observed barrier dysfunction was consequential for skin homeostasis in aged mice, we performed hematoxylin and eosin (H/E) staining of Piezo1-eKO aged mice. Consistent with the reduced barrier function in aged mice, H/E-stained epidermal sections of aged Piezo1-eKO mice exhibited epidermal hyperthickening (Fig. 5A). The hyperthickening was associated with hyperproliferation of the basal cells within the epidermis, whereas no hyperproliferation was detected in young mice (Fig. 5B; Fig. S5A). As the hyperthickening and hyperproliferation were only observed in aged basal cells, we hypothesized that the observed abnormal proliferation was secondary to the barrier function rather than a cell autonomous phenotype. In support of this notion, cultured Piezo1-eKO keratinocytes did not display proliferation abnormalities (Fig. S5B), indicative of a cell non-autonomous effect.

An impairment in skin barrier might trigger an inflammatory response in resident immune cells (Djalilian et al., 2006). To

Journal of Cell Science

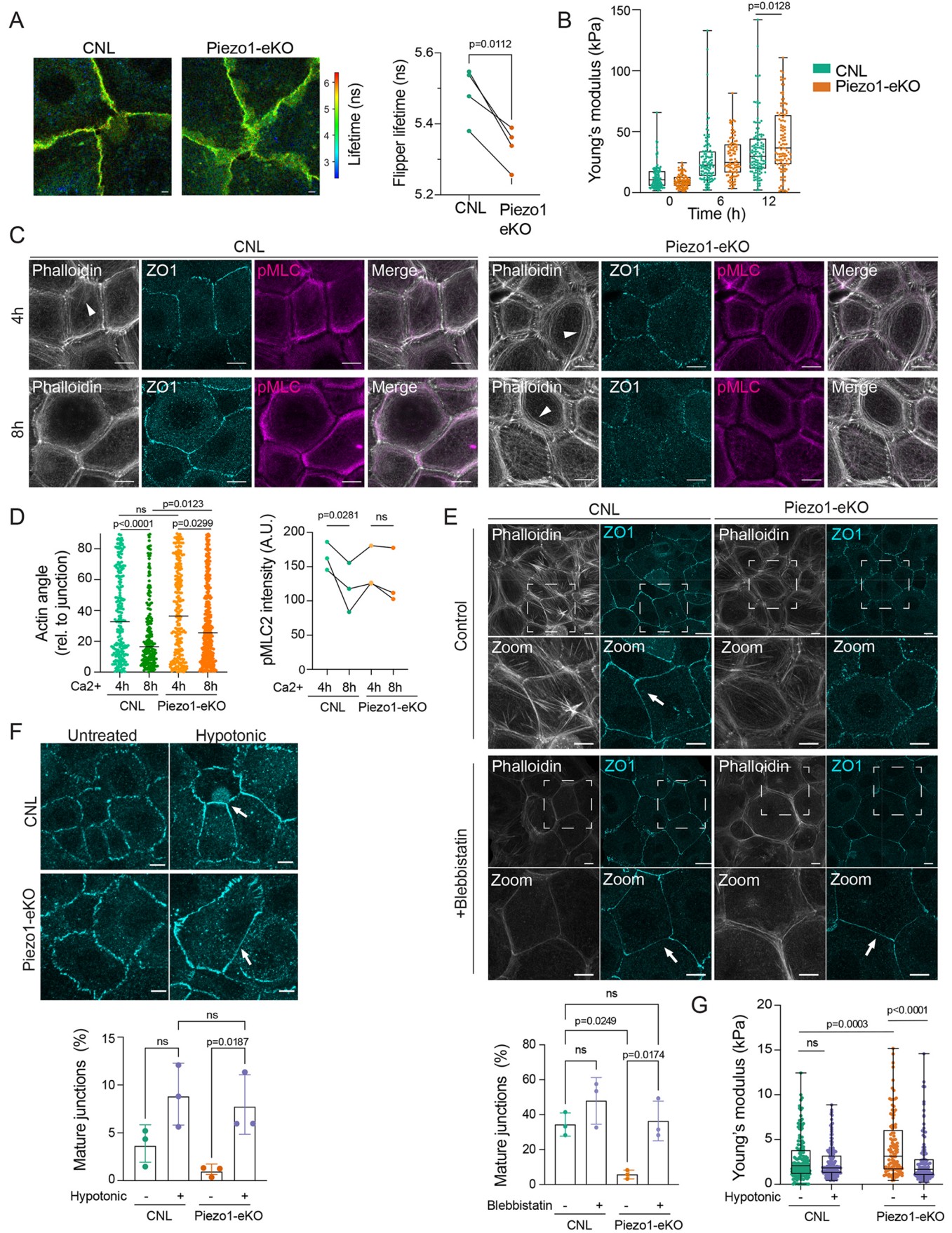

**Fig. 3.** See next page for legend.

**Fig. 3. Requirement of Piezo1 for cell–cell junction maturation can be bypassed by modulating membrane and cortical tension.**
(A) Representative images and quantification of fluorescence lifetime imaging of FLIPPER-TR from CNL and Piezo1-eKO primary keratinocytes. Note decrease in FLIPPER-TR lifetime indicative of lower membrane tension in Piezo1-eKO ($n$=4 independent experiments with ≥40 membrane measurements per condition; paired two-tailed $t$-test). Scale bars: 10 µm. (B) AFM force indentation spectroscopy of CNL and Piezo1-eKO primary keratinocytes before (0 h) and after $Ca^{2+}$ switch. Note elevated elastic modulus in Piezo1-eKO at 12 h ($n$>88 force curves per condition pooled across three independent experiments; Kolmogorov–Smirnov test). (C,D) Representative images (C) and quantification (D) of apical surface projections of ZO-1, pMLC2 and phalloidin-stained primary keratinocytes fixed at 4 h and 8 h post $Ca^{2+}$ switch. Note delay in disassembling thin apical actin stress fibers (arrowheads) in Piezo1-eKO ($n$=3 independent experiments), with >40 cells per experiment percondition; Kruskal–Wallis/Dunn's (left panel), repeated measures ANOVA with Sidak's test (right panel). Scale bars: 10 µm. (E) Representative images and quantification of ZO-1-stained CNL and Piezo1-eKO primary keratinocytes treated with 5 µM blebbistatin for 6 h starting at 6 h of $Ca^{2+}$ switch. Quantification shows faster maturation (arrows) of ZO-1-based cell–cell junctions in blebbistatin-treated Piezo1-eKO cells ($n$=3 independent experiments with >100 cells per experiment per condition; one-way ANOVA with Tukey's test; ns, not significant). Scale bars: 10 µm. (F) Representative images and quantification of ZO-1-stained CNL and Piezo1-eKO primary keratinocytes treated with hypotonic media (140 mOsm). Quantification shows accelerated maturation of ZO-1 based cell–cell junctions (arrows) in Piezo1-eKO cells in hypotonic conditions ($n$=3 independent experiments with >150 cells per experiment per condition; one-way ANOVA with Tukey's test; ns, not significant). Scale bars: 10 µm. (G) AFM force indentation spectroscopy of CNL and Piezo1-eKO primary keratinocytes after 8 h $Ca^{2+}$ switch, with or without treatment with hypotonic medium for 30 min. Note elevated elastic modulus in Piezo1-eKO and restoration of this elevation to control levels after hypotonic treatment ($n$>100 force curves per condition pooled across three independent experiments; Kruskal–Wallis test with Dunn's). For box plots in B and G, the box represents the 25–75th percentiles, and the median is indicated. The whiskers show the minimum-to-maximum range. Results in E, F are mean±s.d. A.U., arbitrary units.

investigate whether this is the case, we immunostained skin sections of aged Piezo1-eKO mice with leukocyte, macrophage and T-cell markers (CD45, F4/80, CD68 and CD3). As anticipated, the absence of Piezo1 in the epidermis led to moderately increased levels of resident T-cells and macrophages (Fig. 5C; Fig. S5C). As further indication of barrier dysfunction, we found that Piezo1-eKO mice exhibited elevated levels of epidermal stress as quantified by increased levels of the stress-associated keratin-6, which is normally not expressed in healthy epidermis (Zhang et al., 2019) (Fig. 5D). Together, these results indicate that Piezo1-eKO mice exhibit a mild age-dependent barrier defect accompanied by hyperproliferation, subtle inflammation and stress in the epidermis.

Finally, we asked why the TJ phenotype only manifests in aged mice (Fig. 5A–D). We hypothesized that aging-induced tissue-stiffening could increase the demands for mechanosensing to balance cortical contractility and membrane tension to efficiently form TJs (Fiore et al., 2020; Koester et al., 2021). To test this, we experimentally elevated cortical contractility *in vitro* in primary keratinocytes by performing biaxial cell monolayer stretching. Analyses of junctional integrity upon application of mechanical stress revealed that whereas the cell–cell junctions of control monolayers were largely unaffected, Piezo1-eKO monolayers showed more ruptures at junctions, resulting in gaps in the monolayer (Fig. 5E). This indicates that Piezo1 is required for mechanical stability of cell–cell junctions, possibly explaining the appearance of the barrier defect only in the aged, mechanically stiff tissue.

## DISCUSSION

Local changes in the balance of forces between cells, associated with events such as changes in tissue mechanics, are central to epithelial homeostasis (Mao and Wickström, 2024; Wyatt et al., 2016). Our current data suggest that balancing between cortical and membrane tension is required for junction maturation from zipper-like primordial junctions into mature TJ belts. We find that this balancing is mediated by the mechanosensitive ion channel Piezo1 in skin keratinocytes. Although formation of primordial junctions occurs normally in the absence of Piezo1, Piezo1-deficient cells exhibit a strong delay in transitioning these actomyosin-linked zipper-like adhesions into mature adhesion belts. This delay can be bypassed by decreasing actomyosin cortex tension or increasing plasma membrane tension. Interestingly, the adhesion maturation defect becomes relevant for *in vivo* epidermal homeostasis only in aged mice, where an impaired TJ barrier is associated with epidermal hyperproliferation and hyperthickening, epidermal stress and an inflammatory response.

The switch from zipper- to belt-like intercellular junctions is tightly coupled to dramatic reorganization of the actomyosin cytoskeleton. In zippers, actomyosin stress fibers perpendicularly associate with adhesions to generate contractility. Upon formation of belts these radially oriented stress fibers transition into peri-junctional actin bundles, associated with tension build up at the intercellular junction itself (Vaezi et al., 2002; Vasioukhin et al., 2000). Thus, profound reorganization of actomyosin is crucial for this transition, and it is known that this switch is regulated both by local RhoA and Cdc42 activation downstream of the polarity master regulator atypical protein kinase C (aPKC) (Sit and Manser, 2011; Überall et al., 1999). However, what defines the precise timing of activation of these pathways has remained unclear. We propose that Piezo1, which transiently localizes to areas of high contractile forces, focal adhesions and zipper-like adherens junctions, acts as a sensor for local tension build-up to determine the timing of the transition. Consequently, Piezo1-deficient cells display prolongation of the zipper-like state. In this respect it is interesting to note that Piezo channels have been implicated in RhoA activity regulation in other mechanosensory processes such as focal adhesion maturation in mammalian cell lines or TJ repair in *Drosophila*, providing a potential downstream mechanism for this effect (Pardo-Pastor et al., 2018; Varadarajan et al., 2022; Yao et al., 2022). Consistent with this, inhibition of ROCK activity rescues timely formation of belt-like junctions also in keratinocytes. Importantly, although there is a delay in maturation of zipper-like primordial junctions into mature belt-like junctions, these junctions will eventually form. Here, compensatory mechanisms such as activity of Piezo2 could be involved. However, whereas morphologically continuous belts are observed, these junctions are mechanically unstable, as indicated by the rupture seen in Piezo1-deficient monolayers upon mechanical stretch.

Although the role of dynamic actomyosin regulation in junction maturation is quite well understood, the role of membrane tension has been elusive. Membrane tension is defined as the force needed for the in-plane deformation of a unit length of membrane, and is driven by shifts in the equilibrium distance between phospholipids in the lipid bilayer (Sens and Plastino, 2015). The large size and curved architecture of Piezo1 trimers has been shown to render Piezo1 directly sensitive to lateral plasma membrane tension (Cox et al., 2016; Guo and MacKinnon, 2017; Haselwandter and MacKinnon, 2018; Lewis and Grandl, 2015). In addition to membrane tension, several studies indicate that the cytoskeleton gates activation of Piezo channels, although the direct link between

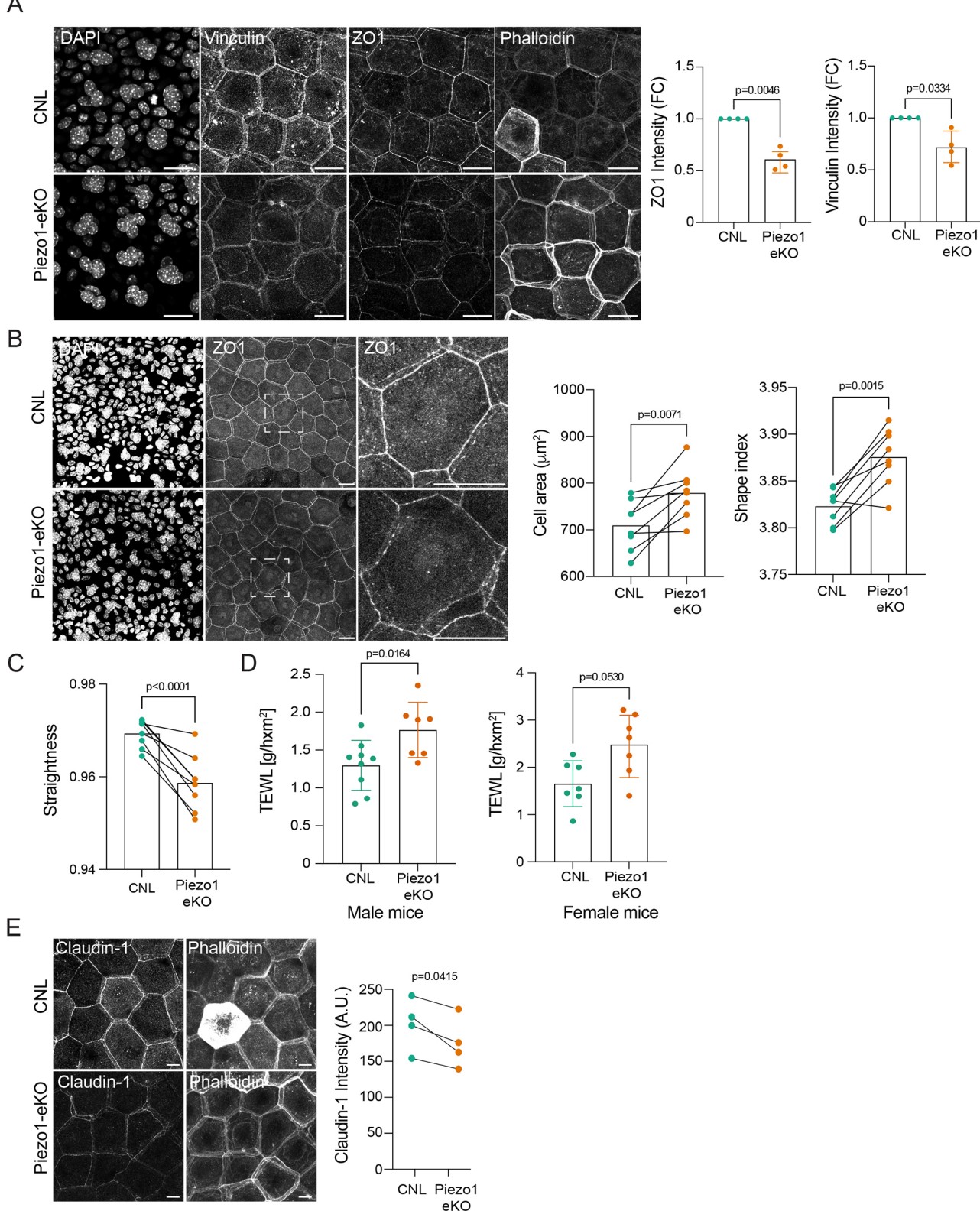

**Fig. 4.** See next page for legend.

Piezo1 and the cortical cytoskeleton is under debate (Bavi et al., 2019; Dumitru et al., 2021; Mylvaganam et al., 2022; Pathak et al., 2014; Verkest et al., 2022; Wang et al., 2022). This dual sensitivity places Piezo1 in an ideal position to integrate membrane tension and cortical tension sensitivity. Interestingly, we observe that Piezo1 is partially located at E-cadherin adherens junctions as well as focal adhesions but is absent from belt-like junctions. This is consistent with previous work showing preferential localization of Piezo1 at

**Fig. 4. Aging leads to defects in TJ maturation and the barrier function of Piezo1-eKO mice.** (A) Representative images and quantification of vinculin, ZO-1 and phalloidin-stained ear whole-mounts from 1-year-old CNL and Piezo1-eKO mice. Note decrease in junctional vinculin and ZO-1 intensity in Piezo1-eKO ($n$=4 mice per genotype; one-sample $t$-test). Scale bars: 20 µm. (B) Representative images and quantification of cell shape from ZO-1-stained ear whole-mount images of CNL and Piezo1-eKO <1-year old mice stained for ZO-1. Dashed rectangle indicates magnified area ($n$=8–10 mice per genotype; paired two-tailed $t$-test). Scale bars: 20 µm. (C) Quantifications of junction straightness ($n$=8–10 mice per genotype; paired two-tailed $t$-test). (D) Quantifications of transepidermal water loss measurements from CNL and Piezo1-eKO 1-year old male and female mice back skin [$n$=9 (CNL male) or 7 mice (other groups); Mann–Whitney]. Scale bars: 20 µm. (E) Representative images and quantification of claudin-1 staining from ear whole-mount images of CNL and Piezo1-eKO <1-year old mice ($n$=4 mice per genotype; paired two-tailed $t$-test). Scale bars: 10 µm. Results in A, D are mean±s.d. A.U., arbitrary units; FC, fold change.

regions of high actomyosin contractility, namely, the contractile rear of migrating keratinocytes where it controlled the speed of wound healing (Holt et al., 2022), within focal adhesions of various cell types (Chen et al., 2018; Yao et al., 2022), and to the intercellular bridge during cytokinesis (Carrillo-Garcia et al., 2021). This leads us to postulate that Piezo1 is localized at areas of high contractility and that it translocates away from the belt-like adhesions.

Notably, the cortical cytoskeleton can control the length scale of membrane tension propagation, thereby indirectly controlling Piezo1 activation via the lipid bilayer (De Belly and Weiner, 2024; De Belly et al., 2023; Ellefsen et al., 2019; Shi et al., 2018, 2022). Interestingly, while sensing membrane tension, Piezo1 has been shown to be involved in a feedback loop where Piezo1 activation leads to elevated membrane tension (Qian et al., 2022). Finally, it has been shown that during a sustained stimulus, Piezo1 currents will decay, preventing them from responding to an additional mechanical stimulus or steady state membrane tension (Coste et al., 2015). These properties could collectively explain our observations that Piezo1 is required for the rapid, well-timed transition of high actomyosin tension-generated zipper-like junctions by localizing transiently specifically at these sites to sense local membrane tension. Once activated, downstream RhoA or Cdc42 signaling would then catalyze the necessary actomyosin rearrangement to generate the mature belt structures, further facilitated by increased membrane tension to bring the plasma membranes together. The reported decay mechanism of Piezo1 activity would subsequently inactivate the channels and the decrease in local actomyosin would facilitate Piezo1 localization away from these structures. Rigorously testing this model is an important task for future work.

Surprisingly, although the phenotype of delayed junction maturation is a robust phenomenon in Piezo1-deficient primary keratinocytes *in vitro*, observed consistently regardless of the age of the mouse from which the cells are isolated from, the *in vivo* phenotype only becomes obvious in aged mice. Tissue culture plastic is orders of magnitudes stiffer than *in vivo* tissues (Discher et al., 2005), including skin, and previous studies have shown that the epidermis becomes stiffer with age (Ichijo et al., 2022; Koester et al., 2021). This would imply that the importance of Piezo1 in local mechanosensing at the junctions becomes essential only at high levels of tissue tension and associated actomyosin contractility. Consistently, reducing contractility *in vitro* facilities timely adhesion maturation in the absence of Piezo1, whereas increasing monolayer tension by stretch makes adhesions less stable.

Collectively these data implicate Piezo1 in sensing membrane tension at sites of high actomyosin contractility, thereby integrating

these two processes to regulate timely adhesion maturation, particularly in tissues that are under high mechanical stress.

## MATERIALS AND METHODS

### Mouse strains

All mouse studies were approved and carried out in accordance with the guidelines of the Finnish national animal experimentation board (ELLA) under permit number ESAVI/22531/2018. C57BL/6J mice with homozygous floxed Piezo1 alleles (Piezo1$^{fl/fl}$, Jackson Labs, stock no. 029213), were bred with K14-Cre mice (Hafner et al., 2004) to achieve Piezo1$^{fl/fl}$-K14-Cre mice with epidermis-specific deletion. No gender differences were detected in phenotypes with the exception of TEWL, so both males and females were analyzed. Piezo1$^{fl/fl}$ and Piezo1$^{fl/wt}$ [collectively termed as control (CNL)] sex-matched littermates were used as control.

### Isolation and culture of primary keratinocytes

Epidermal keratinocytes were isolated from adult (3 months old) control and Piezo1-eKO mice. Following skin isolation, epidermal single-cell suspensions were generated by incubating skin pieces in 0.8% trypsin (Gibco) for 45–50 min. Following trypsinization, the epidermis was separated from dermis by scraping with small forceps. The epidermal cell suspension was pipetted up and down several times and filtered through a 70 µm cell strainer. Cells were resuspended in keratinocyte growth medium [KGM; MEM Spinner's modification (Sigma), 5 µg ml$^{-1}$ insulin (Sigma), 10 ng ml$^{-1}$ EGF (Sigma), 10 µg ml$^{-1}$ transferrin (Sigma), 10 µM phosphoethanolamine (Sigma), 10 µM ethanolamine (Sigma), 0.36 µg ml$^{-1}$ hydrocortisone (Calbiochem), 2 mM glutamine (Gibco), 100 U ml$^{-1}$ penicillin, 100 µg ml$^{-1}$ streptomycin (Gibco), 10% chelated fetal calf serum (Gibco), 5 µM Y27632, 20 ng ml$^{-1}$ mouse recombinant vascular endothelial growth factor and 20 ng ml$^{-1}$ human recombinant fibroblast growth factor-2 (all from Miltenyi Biotec)] as described previously (Chacón-Martínez et al., 2017) and plated on pre-coated [collagen I (10 µg ml$^{-1}$ Millipore), fibronectin (100 µg ml$^{-1}$ Millipore)] culture dishes.

### Transfection and drug treatments

hPiezo1-3xFLAG was from Genecopeia (EXZ6777-Lv181). ZO-1–mEmerald was Addgene #54316. For transfections, cultured keratinocytes were trypsinized using 0.5% trypsin (Gibco 15090-046) with 0.5 mM EDTA. 300,000 cells were seeded in eight-well chamber slides (Lab-Tek) and cultured in KGM for 1–2 days until 80–90% confluent. Transfections were performed with Lipofectamine 3000 (Invitrogen L3000015) according to the manufacturer's protocol. After 6–8 h transfection, the medium was switched to defined keratinocyte serum-free medium (KSFM; Gibco 107850-12) supplemented with 200 µM Ca$^{2+}$ to induce the formation of junctions.

For hypotonic treatment, the KSFM was diluted with sterilized milliQ water at a 1:1 ratio leading to an approximate molarity of 130–160 mOsm. Ca$^{2+}$ was added to a 200 µM final concentration. For hypertonic treatment, the KSFM was diluted with 1 M sucrose solution in a 1:1 ratio leading to an approximate molarity of 760–820 mOsm.

For blebbistatin treatment, primary keratinocytes were first cultured in KSFM plus 200 µM Ca$^{2+}$ for 6 h to induce the initial formation of immature junctions. Subsequently, 5 µM blebbistatin (Sigma B0560) was added in the culture medium and cells were cultured for another 6 h.

For ROCK inhibition, the primary keratinocytes were first cultured in KSFM plus 200 µM Ca$^{2+}$ for 6 h to initiate junction formation. Subsequently, 10 µM Y27632 (Focus Biomolecules 10-2301) was added for 2 h prior to imaging.

### Immunofluorescence

Cells were cultured on glass coverslips and fixed in 4% paraformaldehyde (PFA) in phosphate-buffered saline (PBS) for 10 min at 37°C. After rinsing, cells were incubated with blocking solution [3% BSA and 5% normal goat serum (NGS), 0.3% Triton X-100 in PBS] for 30–45 min at room temperature. After blocking, cells were incubated overnight at 4°C with primary antibodies diluted in 1% BSA, 0.3% Triton X-100 in PBS. After

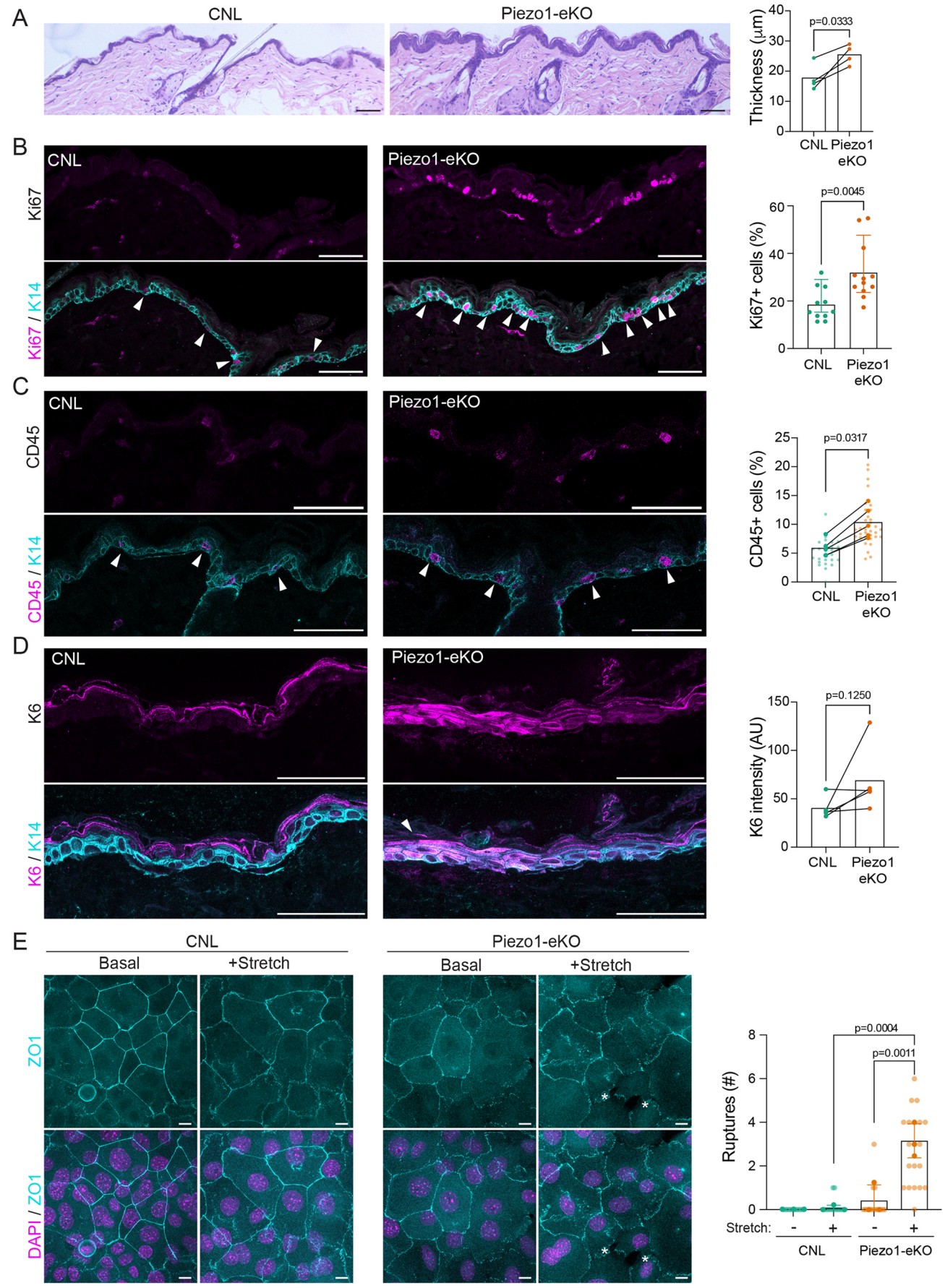

**Fig. 5.** See next page for legend.

**Fig. 5. Deletion of Piezo1 results in impaired epidermal homeostasis and tissue fragility.** (A) Representative images of H/E-stained back skin sections and quantification of epidermal thickness from 1-year-old CNL and Piezo1-eKO back skin ($n$=4 mice per genotype; unpaired two-tailed $t$-test). Scale bars: 100 µm. (B) Representative images and quantification of 1-year-old CNL and Piezo1-eKO mouse back skin stained for Ki-67 (magenta) and keratin-14 (cyan). Note increase in Ki-67-positive cells (arrowheads) in epidermis of Piezo1-eKO mice [$n$=11 (CNL) and 12 (Piezo1-eKO); Mann–Whitney test]. Scale bars: 50 µm. (C) Representative images and quantification of 1-year-old CNL and Piezo1eKO back skin stained for CD45 (magenta) and keratin-14 (K14) (cyan). Note increased number CD45 positive cells (arrowheads) in Piezo1-eKO mice ($n$=5 mice per genotype; Mann–Whitney test). Scale bars: 50 µm. (D) Representative images and quantification of 1-year-old CNL and Piezo1-eKO back skin stained for keratin-6 (K6) (magenta) and keratin-14 (cyan). Note increased expression of keratin 6 (arrowheads) in epidermis of Piezo1-eKO mice ($n$=5 mice per genotype; Mann–Whitney test). Scale bars: 50 µm. (E) Representative images and quantification of ZO-1-stained biaxially stretched CNL and Piezo1-eKO primary keratinocyte monolayers. Note fractures in Piezo1-eKO monolayers (asterisks) upon stretch ($n$=3 independent experiments with >40 cells per experiment per condition; one-way ANOVA with Tukey's test). Scale bars: 10 µm. Results in A–E are mean±s.d. A.U., arbitrary units.

multiple washes in PBS, secondary antibodies were diluted in 1% BSA, 0.3% Triton X-100 in PBS and incubated for 30 min at room temperature. After repeated washes, coverslips were mounted on objective slides with Elvanol mounting medium.

Tissue biopsies were fixed in 4% PFA, dehydrated through graded ethanol series, and embedded in paraffin using standard protocols (Canene-Adams, 2013) and sectioned. Sections were deparaffinized using a graded alcohol series, blocked in 3% BSA and 5% NGS, after which antigens were retrieved using Dako antigen retrieval solution (pH 6 or pH 9). The sections were incubated with primary antibodies diluted in Dako antibody diluent overnight at 4°C. After multiple washes in PBS, secondary antibodies were diluted in 1% BSA, 0.3% Triton X-100 in PBS and incubated for 30 min at room temperature. After washing, slides were mounted in Elvanol.

The following primary antibodies were used in this study: rabbit polyclonal anti-ZO-1 (Invitrogen 40-2200; 1:250), rat monoclonal anti-tension sensitive α18 epitope of α-catenin (Yonemura et al., 2010; kind gift from Akira Nagafuchi, Department of Biology, Nara Medical University, Japan; 1:3000), rabbit polyclonal anti-keratin-6 (Covance PRB-169P; 1:500), mouse monoclonal anti-Vinculin (Millipore MAB3574; 1:250), guinea pig polyclonal anti-keratin K14 (Progen GP-CK14; 1:500), rat monoclonal anti-F4/80 (BMA Biomedicals T-2028; 1:200), mouse monoclonal anti-CD45 (eBioscience 30-F11; 1:200), mouse monoclonal anti-Ki67 (Cell Signaling 9449; 1:400), mouse monoclonal anti-E-cadherin (BD Biosciences 610181; 1:300); mouse monoclonal anti-claudin-1 (Invitrogen 37-4900; 1:100), mouse monoclonal anti-CD3 (17A2) (Invitrogen 14-0032-82) and mouse monoclonal anti-CD68 (KP1, Invitrogen 14-0688-82). The secondary antibodies were conjugated to Alexa Fluor 488, 594 or 647 (Invitrogen; 1:500). Alexa Fluor 647–Phalloidin (Invitrogen A22287; 1:400) was used to stain F-actin. Nuclei were detected with DAPI.

### Preparation of epidermal ear whole-mounts
Epidermal ear whole-mounts were prepared from young and old mice as described previously (Rübsam et al., 2017). Briefly, ears were dissected into dorsal and ventral pieces, cartilage was removed, and the inner (ventral) side of the ear was floated on a 5 µM Dispase II (Sigma D4693) solution at 37°C for 10–15 min. The epidermal layer was peeled off and fixed on ice with ice-cold 4% PFA for 10 min, rinsed in PBS and permeabilized with 0.5% Triton X-100 in PBS for 1 h at room temperature. Following permeabilization, the epidermal sheet was blocked with 3% BSA and 5% NGS after which primary antibodies were diluted in blocking solution and incubated overnight at 4°C. Secondary antibodies, DAPI and phalloidin were subsequently incubated for 45 min at room temperature, after which tissues were rinsed extensively and mounted on objective slides using Elvanol.

### Microscopy
Confocal images were obtained using Leica TCS SP8X with white light laser and Leica Stellaris 8 FALCON/DLS with 20×0.75 NA, 40×1.30 NA and 63×1.40 NA objectives. Live imaging was performed using LSM980 (Zeiss) confocal microscope equipped with an Airyscan2 detector, and an environmental chamber set at 37°C, 5% CO using a 40× immersion objective. Images were acquired with Zeiss ZEN (Zeiss ZEN version 3.5) software where 'joint deconvolution' processing was performed post image acquisition.

### Image processing and analysis
The transition from zipper-like adhesions to mature continuous intercellular junctions was quantified manually using line scans along the junctions. If intensity profiles were continuous, junctions were classified as mature, whereas discontinuous profiles were classified as zipper-like. For quantifying intensities at junctions, maximum projection images were generated, and regions of interest (ROIs) masks were generated using ZO-1-positive junctions. Intensity profiles for junctional stainings were extracted from these masks. For cell shape, ROIs were generated for cell outlines using ZO-1 masks, the area and shape were extracted, and cell shape was calculated by shape index (perimeter/$\sqrt{\text{area}}$). For quantifying junctional length and straightness, ROIs were drawn over ZO-1 junctions manually, length and ferret length values were extracted, and junctional straightness was calculated from ferret length/length. Vinculin staining intensity was calculated by dividing vinculin mean intensity from junctional focal plane with basal focal plane. Colocalization analysis was performed using the Fiji plugin BIOP by segmenting adhesions based on ZO-1 or vinculin and quantifying pixel-by-pixel Pearson's correlation of Piezo–FLAG and ZO-1 or vinculin intensities within these masks (Bolte and Cordelières, 2006).

Orientation of actin at cell–cell junctions was measured with ImageJ and the Orientation-plugin (Püspöki et al., 2016). First, 8 µm-wide ROIs were manually drawn to cover the junction area from projected junctional focal planes from high resolution confocal images. Next, the actin signal was filtered using a 2-pixel median filter to reduce stochastic noise from the scanner. Orientation of actin fibers was then measured from the preprocessed images using Orientation Measure-command (sigma value 2.0). The actin orientation relative to junction orientation was then calculated as smallest angle between the junction orientation and actin descriptive angle for each junction separately.

Quantification of Ki-67-, F4/80- and CD45-positive cells were counted manually. For quantification of keratin 6 intensity in the epidermis, ROIs were generated for the epidermal area and intensities were extracted within these ROIs.

### Laser ablation
For the laser ablation experiments, primary mouse keratinocytes were switched to serum-free medium containing 200 µM of $Ca^{2+}$ and SPY650-FastAct (Cytoskeleton Inc) 8 h prior to the ablation experiment. Imaging and ablation were performed using a Nikon AX R MP microscope equipped with APO LWD 20× water immersion objective (NA 1.0), Chameleon Discovery NX tunable laser (Coherent Corp) and NDD detector with a bandpass filter 604–676 nm to detect SPY650-FastAct. For time-lapse acquisitions SPY650-FastAct was excited at 1040 nm and images were acquired with a frame rate of 620 ms for 60 s after ablation. The ablation was performed at 850 nm with focused line scans on manually defined ROIs drawn over the cell–cell junctions defined by the actin staining using 60% laser output power with 2 µs pixel dwell time. The recoil of the vertices was quantified using ImageJ as described previously, with minor modifications (Liang et al., 2016). The vertices were tracked in time using ImageJ template matching plugin. First the ablation movies were preprocessed using the Noise2Void-algorithm and custom model to improve the signal-to-noise ratio for the tracking (Krull et al., 2019). For this, vertices at both sides of the cell–cell junction were tracked throughout the time-lapse movie and the distance between the vertices and change from initial length was calculated for each time point. Next, unsuccessfully tracked time points were filtered out from the data series and LOWESS smoothed curves were plotted using GraphPad Prism. Finally, the initial recoil values were calculated by fitting non-linear fit in GraphPad Prism.

## PIV

For particle image velocimetry (PIV) analysis cell were stained with SPY650-FastAct (Cytoskeleton Inc) and live imaged at 20 min/frame using a LSM980 confocal microscope and 20x Plan-Apochromat air objective (NA 0.8) starting 2 h after addition of 200uM of Ca2+. Live movies were then deconvolved using the LSM plus deconvolution algorithm and registered using StackReg ImageJ-plugin (Thevenaz et al., 1998). Subsequently, PIV analysis was performed using MATLAB and PIV lab (version 3.10) with custom configuration settings. First, the quality was enhanced using contrast-limited adaptive histogram equalization (CLAHE) with 128 pix window size. Next, the movement vectors were derived using fast Fourier transform- based cross-correlation with four passes (128pix, 64pix, 32pix and 16pix). Vector validation was implemented using image-based validation where low contrast (0.005) and correlation coefficient (0.05) filters were applied. Finally, vector field was completed by interpolating the missing data. Vector fields were then used to calculate movement magnitude per timepoint and mean magnitude per field of view.

## Transepidermal water loss measurement

Transepidermal water loss measuremet (TWEL) was assessed using the noninvasive probe Tewameter TM300 (Courage+Khazaka, Cologne, Germany) as previously described (Bradley et al., 2016; Man et al., 2015) and according to the manufacturer's instructions. The dorsal skin hairs of mice were shaved prior to measure TEWL. Mice were anesthetized using isoflurane during TEWL measurements. TEWL measurements were averaged at 1-s intervals for a 1–2-min period. For consistency, TEWL was measured by placing the probe at the same location on the dorsal skin of the mouse each time and the same person made all TEWL measurements.

## Histological staining

For histological staining of tissue sections, Hematoxylin and Eosin stain kit (Nordic BioSite Cat. # KSC-1ZN4SF-1) was used according to the manufacturer's protocol. After staining, the sections were mounted in Entellan.

## RNA *in situ* hybridization combined with immunofluorescence

RNA *in situ* hybridization was performed using RNAscope technology following manufactured instructions (ACD, Multiplex Fluorescent v2 Assay combined with Immunofluorescence, Integrated Co-Detection Workflow). Briefly, tissue sections (5 µm thickness) were deparaffinized in xylene, followed by dehydration in 100% ethanol and incubated in hydrogen peroxide for 10 min at room temperature (RT) and placed in RNAscope Co-Detection Antigen Retrieval solution (ACD; #323165) for 15 min at 100°C in a pressure cooker. Primary antibodies were incubated overnight at 4°C in a humidified chamber. Post fixation in 4% PFA was performed for 30 min at room temperature, after which RNAscope Protease Plus (ACD; #322381) was added to each sample and incubated at 40°C for 30 min. The Piezo1-RNAscope probe (ACD; #500511-C2) was hybridized following manufacturer instructions using RNAscope Multiplex fluorescent v2 assay (ACD; #323110) followed by detection with TSA plus Fluorescein (Akoya) fluorophore. All the steps were performed in an Hyb-EZ II oven (ACD). After hybridization, the samples were incubated with secondary antibodies for 30 min at room temperature (conjugated to Alexa Fluor 594 or 647; Invitrogen; 1:500). The samples were mounted in Elvanol and imaged using a Zeiss LSM 980 confocal, with Zeiss ZEN (Zeiss ZEN version 3.5) software with a 40× water immersion objective.

Piezo1 RNAscope images were quantified by drawing region of interest mask based on keratin-14-positive or keratin-10-positive cells after fluorescent foci of Piezo1 mRNA were quantified using the Spot Counter tool in Fiji software.

## Force indentation spectroscopy

AFM measurements were performed on cell monolayers plated on glass bottom dishes using a JPK NanoWizard 2 (Bruker Nano) atomic force microscope mounted on a Nikon Eclipse Ti inverted microscope and operated with JPK SPM Control Software v5. Spherical Silicon Nitride 5 µm cantilevers (MLCT, Bruker Daltonics) with a nominal spring constant of 0.065 $Nm^{-1}$ were used for the nanoindentation experiments of the apical surface of cells. For all indentation experiments, forces of up to 9 nN were applied, and the velocities of cantilever approach and retraction were

kept constant at 10 µm $s^{-1}$. All analyses were performed with JPK Data Processing Software (Bruker Nano). Prior to fitting the Hertz model corrected by the tip geometry to obtain Young's Modulus (Poisson's ratio of 0.5), the offset was removed from the baseline, the contact point was identified and cantilever bending was subtracted from all force curves.

## Membrane tension measurements

FLIPPER-TR fluorescent tension probe (Spirochrome, SC020) was utilized to quantify membrane tension (Colom et al., 2018). Cells were cultured on glass bottom dishes and switched to KSFM plus 200 µM $Ca^{2+}$ for 3 h, 4 h or 8 h. FLIPPER-TR (1 µM) was applied 10–15 min prior to measurements. FLIM imaging was performed using a Leica Stellaris confocal microscope equipped with a FLIM module. Leica FALCON/FLIM software was used to record the data in photon-counting mode using a HC PL APO CS2 63×1.40 NA oil immersion objective. Excitation was performed using a pulsed 488 nm laser operating at 20 MHz with emission collected through bandpass 575/625 nm with a HyD X2 detector. Leica FLIM software was used to fit fluorescence decay data from regions of interest to a two-exponential model. The longest lifetime with the higher fit amplitude was used to quantify membrane tension.

## Mechanical stretching

For biaxial stretch, culture plates with a silicon elastomer membrane (Bioflex; FlexCell International Corporation) were coated with fibronectin (10 µg/ml) in PBS at 37°C overnight prior to cell seeding. 1.5 M cells per elastomer were seeded 24-48 h prior to experiment start to obtain 100% confluency, and 12 h before initiation of stretch culture medium was replaced by KSFM plus 200 µM $Ca^{2+}$. Cells were then exposed to cyclic mechanical strain using the Flexcell Tension System (FX4000; FlexCell International Corporation) at 20% elongation, 0.6 Hz frequency for 30 min. After stress cessation, the cells were immediately fixed in culture medium using 8% PFA and immunostained as described above.

## qPCR

RNA was isolated using the Nucleospin RNA Plus kit (Macherey&Nagel), after which cDNA was synthesized using the iScript cDNA synthesis kit (Bio Rad). Quantitative (q)PCR was performed on the StepOne Plus Real Time PCR System (Applied Biosystems) using the PowerUp SYBR Green Master Mix (Applied biosystems). Gene expression changes were calculated following normalization to GAPDH using the comparative Ct (cycle threshold) method.

The primers used were: Piezo1 forward 5′-AGCATACCAGGTCACACAGGTC-3′, reverse 5′-CCAAAGGCTACCGTTTTGTCCC-3′ and GAPDH forward 5′-GGTGTGAACGGATTTGGCCGTATTG-3′ reverse 5′-CCGTTGAATTTGCCGTGAGTGGAGT-3′.

## Statistics and reproducibility

Statistical analyses were performed using GraphPad Prism software (GraphPad, version 10). Statistical significance was determined by the specific tests indicated in the corresponding figure legends. Only two-tailed tests were used. In all cases where a test for normally distributed data was used, normal distribution was confirmed with the Kolmogorov–Smirnov test ($\alpha$=0.05). All experiments presented in the manuscript were repeated at least in three independent biological replicates. Experimental groups were based on mouse phenotypes, and treatments were applied to primary cells isolated from individual mice per genotype per experiment. No statistical methods were used to predetermine sample sizes; they were based on experience with the methodology. Researchers were aware of experimental conditions in all analyses. No datapoints were removed with the exception of AFM measurements, where outlier identification was carried out to remove rare individual measurements that represented apparent artefacts arising from contribution of substrate to the measurements.

## Acknowledgements

We thank Karolina Punovuori for support with experiments, Hanne Ahola and Claudia Ortmeier for expert technical assistance, and the Max Planck Institute BioOptics and Biomedicum Imaging Unit, HiLIFE for support with imaging.

**Competing interests**
The authors declare no competing or financial interests.

**Author contributions**
Conceptualization: C.M.N., S.A.W.; Formal analysis: A.J., A.S., S.-M.M., S.A.W.; Investigation: A.J., C.V., S.-M.M., F.P., L.C.B.; Methodology: A.S., C.V., S.-M.M., M.R., L.C.B., S.A.W.; Project administration: S.A.W.; Resources: C.M.N., S.A.W.; Supervision: C.M.N., L.C.B., S.A.W.; Validation: A.J.; Visualization: A.J., S.A.W.; Writing – original draft: A.J., S.A.W.; Writing – review & editing: A.S., C.V., M.R., C.M.N., L.C.B.

**Funding**
This work was supported by a Doctoral Programme in Biomedicine (DPBM) University of Helsinki, Biomedicum Young Investigator grant (both to A.J.), Instrumentarium Science Foundation and Academy of Finland postdoctoral fellowships (to A.S.), European Union's Horizon 2020 research and innovation programme under Marie Skłodowska-Curie grant agreement no. 101032331 (to C.V.), Academy of Finland Research Fellowship 332821 (to L.C.B.), the Sigrid Juselius Foundation, Helsinki Institute of Life Science, and Academy of Finland Center of Excellence BarrierForce (346131) and R'Life Programme consortium NucleoMech (317597) and the Max Planck Society (all to S.A.W.). Open Access funding provided by Max-Planck-Gesellschaft. Deposited in PMC for immediate release.

**Data and resource availability**
All analysis scripts and data that support the conclusions are available from the authors on request.

**First Person**
This article has an associated First Person interview with the first author of the paper.

**Peer review history**
The peer review history is available online at https://journals.biologists.com/jcs/lookup/doi/10.1242/jcs.263938.reviewer-comments.pdf

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
