## [Peer Review File · Journal of Cell Science]

Piezo1 balances membrane and cortex tension to stabilize intercellular junctions and maintain the epithelial barrier

Ahsan Javed, Aki Stubb, Clémentine Villeneuve, Satu-Marja Myllymäki, Franziska Peters, Matthias Rübsam, Carien M. Niessen, Leah C. Biggs and Sara A. Wickström
DOI: 10.1242/jcs.263938

Editor: Andrew Ewald

Review timeline

Submission to Review Commons:	21 November 2024
Submission to Journal of Cell Science:	19 February 2025
Editorial decision:	24 February 2025
First revision received:	10 July 2025
Accepted:	24 July 2025

Reviewer 1

Evidence, reproducibility and clarity

The studies described in this manuscript investigated the mechanical regulation of tight junction (TJ) maturation in the epidermis using a combination of in vitro and in vivo analysis. The findings indicate that during calcium-induced cell-cell adhesion in keratinocytes, there is an initial build up cortical tension in the actin cytoskeleton, followed by an increase in membrane tension, which is required for formation of mature TJs. The studies also demonstrate that loss of Piezo1 delays TJ maturation via defects in membrane tension. Loss of Piezo1 also impaired epidermal homeostasis and barrier function in aged mice. The authors propose that the balance in forces between the cortex and membrane is essential for TJ assembly and is mediated by Piezo1.

Overall, the studies are carefully designed and executed and provide a clear role for membrane tension and Piezo1 in TJ development, making use of molecular forces sensors, imaging, and chemical and genetic perturbations. However, not all of the conclusions are fully supported by the data, and some key findings require additional quantitative and statistical analysis.

1. The statement at the end of page 5 ("This indicated that formation of belt-like...") is somewhat overinterpreted from the data shown. To draw conclusions about a switch to reduced contractility at adhesions requires more careful spatio-temporal quantification of F- actin and pmyosin beyond the example single cells shown in 1b. It would also help to see the localization of Ecadherin during this process.
2. To avoid confusion, the authors should pay careful attention to terminology and be specific when referring to adherens junctions or TJs, rather than just junctions generally.
3. The labelling of Figure 2b could be clearer. Were the CNL cells also transfected with Piezo1 or mock transfected to control for general effects of transfection? This was not clear from the figure captions.
4. In Figure 2c-g it is not specified which timepoints the images represent, and the qualitative description of changes in localisation require quantification.
5. The importance of Piezo1 in junction maturation is somewhat overstated throughout. While Piezo KO clearly delays TJ maturation, the process can still be completed. In the absence of Piezo1 what triggers the rise in membrane tension? Could there be any compensation from Piezo2?
6. Some of the differences noted are subtle and not strongly significant, such as K6expression, Ca⁺⁺ induced Piezo1 expression, and F4/80 staining. The conclusions related

to these responses should be tempered or qualified.

7. Analysis of the immune infiltration and the suggested inflammatory response in aged mice is fairly preliminary and not well supported by the data. A second marker of macrophages and addition of T cell markers would help clarify the type of immune response. It would also help to describe the localisation of specific immune cells in more detail and include a direct marker of inflammation (e.g. inflammatory cytokines).

8. OPTIONAL: Although not essential for the conclusions of the study, the impact and insight could be improved by providing more analysis of the mechanism for the role of Piezo1. For example, does the build up of cortical tension trigger changes in ion channel signalling, and how does this then regulate membrane tension? Is RhoA or aPKC involved?

Significance

The process by which epithelia assemble and maintain effective barriers is complex and requires precise spatio-temporal regulation. This study provides some new insight into the mechanical regulation of TJ assembly within the epidermis. It builds upon previous work that identified essential biomechanical cross-talk between adherens junctions and TJs and adds some new information on the timings and specific roles of membrane tension and Piezo1. The interplay between cortical and membrane tension is noteworthy, and this mechanism may have important implications in other barrier tissues. A limitation of the study is a lack of mechanistic detail in how the mechanical switch occurs during TJ maturation, including the specific molecules, structures, and interactions with Piezo1.

The study also describes the functional implications, whereby loss of Piezo1 in the mouse disrupts barrier integrity. However, these effects were quite subtle. Barrier homeostasis was only disrupted in aged mice, and in vitro, loss of Piezo1 delayed but did not prevent junction maturation. It is therefore interesting to speculate what other mechanisms may be involved in TJ maturation. A potential limitation here is also a lack of detail in the analysis of the inflammatory and immune response in Piezo KO skin.

Reviewer 2

Evidence, reproducibility and clarity

This manuscript describes the role of the mechanosensitive ion channel Piezo1 in epithelial junction assembly, using Piezo-1-KO primary epidermal keratinocytes in vitro and mouse skin in vivo. The authors conclude that Piezo1 allows balancing of membrane versus cortical tension to stabilize junctions and promote tight junction (TJ) barrier integrity assembly. The conclusion that Piezo1 has an important function in the formation and maintenance of apical junctions of keratinocytes both in vitro and in vivo is well documented by experiments in WT, KO and rescue cells/tissues where different parameters are carefully measured: protein localization, quantification of mature junctions, membrane tension using the flipper probe, use of the myosin inhibitor blebbistatin, analysis of cortical stiffness by AFM, etc. Although, the physiological relevance and the mechanism through which Piezo operates in young skin are not clear, the authors make reasonable claims, that are not too speculative.

Major comments:

1. The Supplementary Figure 4d (panel d) that is described in the Results section is missing. It supposedly shows that 1 year-old Piezo1-eKO mice display an increase in transepidermal water loss, indicating that TJ barrier function is compromised. The Figure legend for the panel is also missing. Please provide the Figure panel and the legend.
2. TJ barrier function depends on claudins, and the loss of claudin-1 leads to transepidermal water loss (please cite the relevant paper from the Tsukita lab). Considering that altered TJ barrier function is observed only in 1-yr old mice (Supplementary Figure to be shown, see point n.1) and not in young mice (Suppl. Fig. 3f-h), the expression pattern of the main claudin isoforms, and especially claudin-1, in the different cell populations (see Suppl. Fig. 3b, or by IF analysis) in young vs old and WT vs KO mice must be provided, to provide a mechanistic basis for the observed TJ

barrier phenotype. This would help to determine if the phenotype is linked to altered claudin expression or to altered (increased) perijunctional tension.

3. Mechanistically, the authors mention in the Discussion that Piezo1 might act through RhoA signaling. In Rüksam et al 2017 the authors showed that the uppermost viable layer of the skin has increased apical junctional tension, due to anisotropy of AJ distribution which correlates with EGFR activation and localization. In this context, it is important to know if KO of Piezo-1 affects EGFR localization and signaling, and to probe the RhoA pathway using for example the ROCK inhibitor, instead of blebbistatin.

Minor comments:

1. The Methods sections should be improved with additional details. For example, the description of quantification of junctional labeling is vague, and there is often no or little indication in the Legends that specifies number of experiments and junctional segments. In addition, quantification of junctional stainings for specific proteins should be done using a junctional reference marker and not as "absolute" values, because there can be variability of staining between samples and experiments. This is especially important when measuring ZO-1, which is a dual AJ-TJ protein (for example at zipper-like junctions ZO-1 colocalizes with AJ markers). Double labelling with a true TJ marker (occludin or cingulin) and/or a true AJ marker (PLEKHA7, afadin, Ecadherin or a catenin) and quantifying junctional labeling by ratio is highly recommended. This is particularly important when evaluating tension-sensitive epitopes/antigens (alpha-catenin, vinculin, etc)
2. Please use ZO-1 (and ZO-2) consistently, instead of ZO1 (or ZO2), which is completely inaccurate.
3. Please cite Furuse et al 2002 JCB (see above).
4. Please include statistical data in Figure Legends, specifying the number of separate experiments and number of samples. At least three experiments is recommended.
5. At the end of the introduction the authors mention "putative" occludin-containing TJs. I would delete putative. Epithelial junctions that contain a continuous circumferential linear distribution of occludin/ZO-1/cingulin and form a barrier comply with the definition of a TJs (Citi et al JCS 2024) .
6. Please insert page numbers in the manuscript.

Significance

The notion that mechanosensitive calcium channels contribute to the formation of continuous apical junctions (repair and assembly) was introduced by the Miller lab, using *Xenopus* oocytes. This manuscript provides a significant conceptual advance, not only by using in vitro and in vivo mouse (mammalian) epidermal keratinocytes as model system, but especially by using Piezo1-KO and rescue experiments, which was not done in the *Xenopus* model.

This research would be of great interest to cell biologists interested in epithelial differentiation, polarization and junction assembly, and to clinicians that are interested in the molecular basis of skin pathophysiology.

My expertise is in the biochemistry, cell biology and mechanobiology of epithelial junctions. I have used *Xenopus* embryos, cultured epithelial cells, primary keratinocytes and keratinocyte cell lines and KO mice as model systems. The research of my group focuses on how specific cytoskeletal proteins are organized to transmit forces and are recruited to junctions, and how junctional proteins respond to mechanical force. I have experience in all of the methods described in this paper, except for transepidermal water loss measurement, in situ RNA hybridization and mechanical stretching experiments.

Reviewer 3

Evidence, reproducibility and clarity

This manuscript addresses the important topic of cell-cell junction maturation and mechanical stability, with a specific focus on how mechanotransduction through the Piezo1 channel regulates

these processes. The authors present compelling *in vivo* evidence demonstrating that Piezo1 plays a role in junction stability and barrier function, particularly in aged tissue. The work makes a valuable contribution to our understanding of mechanotransduction in epithelial biology. However, several aspects of the mechanistic model and *in vitro* experiments require additional development to fully support the authors' conclusions.

Major Strengths:

- The *in vivo* experiments are well-designed and provide convincing evidence for Piezo1's role in barrier function
- The study identifies an important connection between mechanical sensing and junction maturation
- The age-dependent phenotype provides interesting insights into tissue mechanics

1. Areas Requiring Additional Development:

a. Mechanistic Model Definition

A major issue is that the central concept of Piezo1 "balancing membrane and cortical tension" requires more precise definition and experimental support. The authors need to clearly explain what this balance means mechanistically and how it is achieved.

b. Localization-Function Discrepancy

There is an important inconsistency between the authors' claims about Piezo1's role and its localization: while they conclude that Piezo1 is crucial for mechanical stability, they also show that Piezo1 is not localized at mature junctions. This apparent contradiction needs to be addressed with a clear mechanistic explanation.

c. Quantification and Statistical Analysis

Several key conclusions would benefit from more rigorous quantification:

- The quantitation of junction maturation in Fig. 1a and 2a should include independent analysis of each experiment rather than pooling cells from multiple experiments
- Actin morphology and pMLC2 levels at junctions in Fig. 1 need systematic quantification
- Cytoskeletal dynamics and morphological changes in Piezo1-eKO cells (Fig. 2a) require quantification

d. Methodological and Timeline Clarity

The analysis methods and temporal aspects of several experiments need better documentation:

Analysis Methods:

The quantification method for mature adhesions (used in Figs. 1a, 1e, 1f, 2a) needs clarification. The Methods section states that "The transition from zipper-like adhesions to mature continuous intercellular junctions were quantified manually," but crucial details are missing:

- What specific criteria defined a "continuous junction"?
- Was this based on complete visibility of the cell perimeter as one junction?
- How were cells classified as having continuous versus zipper-like adhesions?

e. The protein intensity quantification at junctions requires methodological clarification. The Methods state "For quantifying intensities at junctions, max projection images were generated, and region of interests (ROIs) were restricted to ZO1-positive junctions." However:

- Were ROIs drawn empirically by the user? Or was the ZO-1 signal used to make a mask?
- Was there an automated step to determine junctional areas (e.g., intensity threshold)?
- Was the analysis blinded?

If subjective methods were used, this should be clearly stated and potential variability addressed.

2. Timeline Documentation:

For blebbistatin experiments (Fig. 1e), specify observation timeframes and quantify the extent of accelerated maturation

The hypotonic shock experiment (Fig. 3e) timeline needs clarification:

- When were measurements taken relative to Ca²⁺ switch?
- Duration of hypotonic media exposure?
- Were there time-dependent effects in cell response?

3. Data Support and Interpretation

a. Several conclusions require additional support or clarification:

- The claim about "more dynamic cytoskeletal motion and irregularly shaped" cells (Fig. 2a) is not supported by the provided data. Quantification of dynamics and cell shape are needed to support this conclusion. Cytoskeletal imaging data would also be useful.

b. The interpretation of junctional tension requires revision:

- Current conclusions about increased junctional tension are inferred indirectly from vinculin (Fig. 1c) and a18-catenin (Fig. S1a) immunostaining images.
- Consider either:

a) Adding direct junctional tension measurements (e.g., optical measurements, PMID 31964776)

b) Limiting claims to well-supported morphological differences and moving tension-related interpretations to the Discussion as speculative elements

c. The description "Analysis of vinculin translocation to intercellular junctions showed reduced levels of vinculin at cell-cell contacts, but abundant vinculin at cell-matrix adhesions (Supplementary Fig. S2a), indicating abnormal build-up of stresses at intercellular junctions of Piezo1-eKO cells" needs revision:

- "Build-up" suggests higher tensions in Piezo1-eKO cells, which contradicts impaired adhesion maturation findings. Suggest replacing with "distribution" or "organization" "Intercellular" is used ambiguously to include both cell-cell and cell-matrix adhesions

4. Literature Context:

The discussion should incorporate recent relevant literature on Piezo1's role in tight junction regulation (e.g., PMID 37005489, PMID 33636174, PMID 31409093)

5. Technical Considerations

- For localization studies (Fig. 2), using keratinocytes from Piezo1-tdTomato mouse (JAX #029214) would be preferable to heterologously-expressed Piezo1-FLAG, as it would avoid potential artifacts from non-physiological expression levels
- Supp Fig. 1b requires additional replicates
- The Fig. 3A legend states "Note increase in FLIPPER-TR lifetime indicative of elevated membrane tension in Piezo1-eKO" when the data actually shows the opposite - a decrease in Flipper-TR lifetime indicating lower membrane tension

6. Conceptual and Experimental Clarity Needed

Several statements require clearer explanation or additional supporting evidence:

a. Regarding junction maturation mechanisms:

The authors state: "This indicated that formation of belt-like adhesions was associated with initial contractility build-up by actomyosin stress fibers linked to junctions, followed by a switch to parallel actomyosin bundles and reduced contractility at adhesions, while the junctions themselves were stabilized in a stressed state indicated by a strengthened actin- junction link." Each part of this claim needs experimental support:

- The "initial contractility build-up by actomyosin stress fibers linked to junctions" needs to be demonstrated
- The "switch to parallel actomyosin bundles and reduced contractility at adhesions" requires quantification
- The claim about "junctions themselves were stabilized in a stressed state" needs stronger evidence

b. The statement "contact expansion from zippers to a belt requires collaborative regulation of adhesion tension and actomyosin cytoskeleton to lower interfacial tension at the contact" is unclear and needs clarification

c. The claim "Concomitant with emergence of continuous junctions (8h), the stress fibers were replaced by thick actin bundles positioned perpendicularly to junctions (Fig. 1b)" is not clearly supported by the data

7. Regarding experimental interpretation:

- In Fig. 1e, the authors claim that 5 μ M blebbistatin accelerates junction maturation, but this conclusion is not supported by the statistics ($p = 0.0784$). Additionally, the timeframe of observation and the quantification of maturation speed should be specified
- The results section describing Fig. 3 presents seemingly disconnected observations without clear mechanistic links between them, making it difficult to follow the authors' logic and support their conclusions
- The mechanism by which both reduced contractility (blebbistatin) and increased membrane tension can accelerate maturation (Fig. 1e, f; and also in Piezo1-eKO Fig. 3d, e) needs explanation. The fact that these interventions also accelerate maturation also in Piezo1-eKO suggests a mechanism independent of Piezo1 which is at odds with their broad conclusion that Piezo1 balances membrane tension and cortical contractility in the maturation process. The precise mechanism of Piezo1's role in sensing membrane and cortex tension requires clarification.
- How Piezo1 maintains mechanical stability of mature junctions despite not being localized there needs to be explained

8. Suggested Additional Experiments:

a. Optional: Given the age-dependent tissue stiffness effects proposed by the authors, examining keratinocyte behavior in vitro on substrates of varying stiffness would provide valuable insights

b. Optional: Direct measurements of tension at cell-cell junctions where Piezo1 localizes would help validate the proposed mechanical model

9. Minor Points:

- The cell biology sections, particularly descriptions of in vitro experiments, would benefit from a thorough revision to improve precision and clarity. For instance, the Results section describes "Analysis of vinculin translocation to intercellular junctions" when no translocation is actually being studied
- Figure legends should clearly indicate what individual data points represent
- Several conclusions are overstated. For example, the authors conclude that "Piezo1 controls the maturation process" and that "Piezo1 is required for cell junction maturation into junctional belts" based on Fig. 2. These are exaggerated claims since maturation still progresses in Piezo1's absence, just more slowly. "Regulates" or "modulates" would be more appropriate terminology

In conclusion, while this manuscript presents important findings regarding Piezo1's role in junction maturation and stability, addressing the mechanistic and quantification issues outlined above is essential for supporting the authors' conclusions. The authors have laid groundwork for understanding an important biological process, and addressing these points would help readers better appreciate the significance of their findings.

Significance

General Assessment: This study investigates the critical role of mechanosensing in epithelial barrier formation and maintenance, with a particular focus on Piezo1's contribution to junction maturation and stability. The work's primary strengths lie in its compelling in vivo demonstrations of Piezo1's importance for barrier function, particularly in aged tissue, and its identification of a novel connection between mechanical sensing and junction maturation.

The age-dependent phenotype provides valuable insights into tissue mechanics and barrier maintenance. However, the mechanistic understanding of how Piezo1 coordinates these processes requires further development, particularly regarding the proposed balance between membrane and cortical tension.

Advance: This work provides several important advances:

1. First demonstration of Piezo1's role in regulating the maturation of cell-cell junctions from zipper-like to belt-like structures
2. Novel insights into how mechanical forces influence junction maturation through

mechanosensitive ion channels

3. Important connection between aging, tissue mechanics, and barrier function

4. Integration of mechanical sensing with junction assembly and stability

The findings extend our understanding of epithelial barrier formation beyond traditional molecular pathways to include mechanotransduction, suggesting new therapeutic possibilities for barrier dysfunction. The age-dependent phenotype is particularly significant as it reveals how mechanical properties of tissue influence barrier maintenance over time.

Audience: This research will be of broad interest to multiple communities:

- Cell biologists studying junction assembly and epithelial organization
- Mechanobiologists interested in force transmission and sensing
- Ion channel researchers interested in the physiological roles of channels
- Aging researchers investigating tissue barrier function
- Bioengineers developing therapeutic strategies for epithelial barriers

The findings have both basic research and translational implications, particularly for understanding and treating age-related barrier dysfunction in epithelia.

Reviewer Expertise: Cell biology, mechanobiology, live cell imaging, quantitative image analysis, ion channels

I have sufficient expertise to evaluate all aspects of the manuscript except for the specific age-related physiological changes in mouse skin, which falls outside my area of expertise.

Author response to reviewers' comments

Manuscript number: RC-2024-02799

Corresponding author(s): Sara A. Wickström

1. General Statements [optional]

We thank the reviewers for the overall very positive assessment of our work and finding the study carefully executed and representing an important advance in the field. We are also grateful for the expert suggestions that will help us to further strengthen the manuscript. As outlined in detail below, we are able to address the reviewer comments in full. We will perform a panel of additional mechanical measurements as well as in vivo characterization of the Piezo1 null mouse phenotype to more clearly elucidate the proposed novel mechanism of balancing membrane tension and cortex tension in the timely formation of mature tight junctions, and the role of Piezo1 in controlling this balance. In addition, we will carefully edit the manuscript text to provide more details and clarify conclusions, as requested by the reviewers.

2. Description of the planned revisions

Reviewer #1

Evidence, reproducibility and clarity

The studies described in this manuscript investigated the mechanical regulation of tight junction (TJ) maturation in the epidermis using a combination of in vitro and in vivo analysis. The findings indicate that during calcium-induced cell-cell adhesion in keratinocytes, there is an initial build up cortical tension in the actin cytoskeleton, followed by an increase in membrane tension, which is required for formation of mature TJs. The studies also demonstrate that loss of Piezo1 delays TJ maturation via defects in membrane tension. Loss of Piezo1 also impaired epidermal homeostasis

and barrier function in aged mice. The authors propose that the balance in forces between the cortex and membrane is essential for TJ assembly and is mediated by Piezo1.

Overall, the studies are carefully designed and executed and provide a clear role for membrane tension and Piezo1 in TJ development, making use of molecular forces sensors, imaging, and chemical and genetic perturbations. However, not all of the conclusions are fully supported by the data, and some key findings require additional quantitative and statistical analysis.

We thank the reviewer for this positive assessment of our work and finding the study carefully done and executed. We are also grateful for the expert suggestions that helped us to further strengthen the manuscript.

1. The statement at the end of page 5 ("This indicated that formation of belt-like...") is somewhat overinterpreted from the data shown. To draw conclusions about a switch to reduced contractility at adhesions requires more careful spatio-temporal quantification of F-actin and myosin beyond the example single cells shown in 1b. It would also help to see the localization of Ecadherin during this process.

We will add quantifications of F-Actin and myosin, higher magnification zoom-ins, and include an E-cadherin stain to more clearly demonstrate this finding. In addition, we will perform live imaging of F-actin to better capture the dynamic changes.

2. To avoid confusion, the authors should pay careful attention to terminology and be specific when referring to adherens junctions or TJs, rather than just junctions generally.

We apologize for the lack of clarity and will carefully edit the text to make clear where we refer to junctions in general rather than adherens or tight junctions specifically.

3. The labelling of Figure 2b could be clearer. Were the CNL cells also transfected with Piezo1 or mock transfected to control for general effects of transfection? This was not clear from the figure captions.

We had also transfected CNL cells as control but did not include this data in the manuscript. We will include this data in the revised manuscript to demonstrate that the effects are not due to transfection artefacts.

4. In Figure 2c-g it is not specified which timepoints the images represent, and the qualitative description of changes in localisation require quantification.

We will include time point information and quantifications.

5. The importance of Piezo1 in junction maturation is somewhat overstated throughout. While Piezo KO clearly delays TJ maturation, the process can still be completed. In the absence of Piezo1 what triggers the rise in membrane tension? Could there be any compensation from Piezo2?

This is precisely the conclusion we have intended to convey - Piezo1 regulates the timely maturation of junctions but in the end these junctions are formed. However, the mechanical stretch experiments in Fig. 5E clearly show that if exposed to extrinsic stress, these junctions are not mechanically stable. Further, the KO mice display a clear barrier defect, showing that lack of Piezo1 compromises the function of the junctions. We will carefully edit the text to avoid overstatement.

We propose that rather than being a key determinate of absolute membrane tension, Piezo1 is required to adjust cell surface tension (cortical tension and membrane tension) to allow two adjacent cells to generate a continuous, stable interface that allows junction formation. In this scenario there is no "compensation" from an alternative pathway, the process will still occur, it is simply less efficient. We will clarify this conclusion in the discussion section.

Importantly, in the epidermis, Piezo2 is not expressed in the epidermal keratinocytes but its expression is restricted to the specialized sensory organ keratinocytes, the Merkel cells (Woo et al., Nature 2014; PMID 24717433). We further addressed if Piezo2 becomes upregulated in the Piezo1 KO but did not observe this, so we reasoned that it is unlikely that Piezo2 is compensating, although we have not formally ruled that out. We will add this aspect to the discussion.

6. Some of the differences noted are subtle and not strongly significant, such as K6 expression, Ca⁺⁺ induced Piezo1 expression, and F4/80 staining. The conclusions related to these responses should be tempered or qualified.

We have performed new stainings for additional macrophage and T-cell markers to strengthen these conclusions. We will include this new data in the manuscript. We will further carefully edit the manuscript to avoid overstating any conclusions.

7. Analysis of the immune infiltration and the suggested inflammatory response in aged mice is fairly preliminary and not well supported by the data. A second marker of macrophages and addition of T cell markers would help clarify the type of immune response. It would also help to describe the localisation of specific immune cells in more detail and include a direct marker of inflammation (e.g. inflammatory cytokines).

As mentioned above, we have performed new stainings for additional macrophage and T-cell markers to strengthen these conclusions. We will include this new data in the manuscript.

8. OPTIONAL: Although not essential for the conclusions of the study, the impact and insight could be improved by providing more analysis of the mechanism for the role of Piezo1. For example, does the build up of cortical tension trigger changes in ion channel signalling, and how does this then regulate membrane tension? Is RhoA or aPKC involved?

We appreciate these suggestions. Due to the differences in time scales - adhesion maturation takes hours vs ion channel signaling as visualized with for example Calcium reporters occurs in seconds, it has been challenging to combine these two readouts to understand the relationship between ion channel activity, cortical tension build-up and adhesion maturation. Calcium imaging requires frame rates in the second time scale, which precludes long term imaging due to phototoxicity. We will analyze the involvement of RhoA activity by measuring activity level differences in wild type and Piezo1-deficient cells. We will further test the potential of RhoA inhibition to rescue Piezo1 mutant phenotype and activity measurements (RhoA Elisa).

Significance

The process by which epithelia assemble and maintain effective barriers is complex and requires precise spatio-temporal regulation. This study provides some new insight into the mechanical regulation of TJ assembly within the epidermis. It builds upon previous work that identified essential biomechanical cross-talk between adherens junctions and TJs and adds some new information on the timings and specific roles of membrane tension and Piezo1. The interplay between cortical and membrane tension is noteworthy, and this mechanism may have important implications in other barrier tissues. A limitation of the study is a lack of mechanistic detail in how the mechanical switch occurs during TJ maturation, including the specific molecules, structures, and interactions with Piezo1.

The study also describes the functional implications, whereby loss of Piezo1 in the mouse disrupts barrier integrity. However, these effects were quite subtle. Barrier homeostasis was only disrupted in aged mice, and in vitro, loss of Piezo1 delayed but did not prevent junction maturation. It is therefore interesting to speculate what other mechanisms may be involved in TJ maturation. A potential limitation here is also a lack of detail in the analysis of the inflammatory and immune response in Piezo KO skin.

Reviewer #2

Evidence, reproducibility and clarity

This manuscript describes the role of the mechanosensitive ion channel Piezo1 in epithelial junction assembly, using Piezo-1-KO primary epidermal keratinocytes in vitro and mouse skin in vivo. The authors conclude that Piezo1 allows balancing of membrane versus cortical tension to stabilize junctions and promote tight junction (TJ) barrier integrity assembly. The conclusion that Piezo1 has an important function in the formation and maintenance of apical junctions of keratinocytes both in vitro and in vivo is well documented by experiments in WT, KO and rescue cells/tissues where different parameters are carefully measured: protein localization, quantification of mature junctions, membrane tension using the flipper probe, use of the myosin inhibitor blebbistatin, analysis of cortical stiffness by AFM, etc. Although, the physiological

relevance and the mechanism through which Piezo operates in young skin are not clear, the authors make reasonable claims, that are not too speculative.

We thank the reviewer for this positive assessment of our work and finding the findings well documented. We are also grateful for the expert suggestions that helped us to further strengthen the manuscript.

Major comments:

1. The Supplementary Figure 4d (panel d) that is described in the Results section is missing. It supposedly shows that 1 year-old Piezo1-eKO mice display an increase in transepidermal water loss, indicating that TJ barrier function is compromised. The Figure legend for the panel is also missing. Please provide the Figure panel and the legend.

The reviewer might have confused the figure labeling, this data is shown in the main Figure 4d.

2. TJ barrier function depends on claudins, and the loss of claudin-1 leads to transepidermal water loss (please cite the relevant paper from the Tsukita lab). Considering that altered TJ barrier function is observed only in 1-yr old mice (Supplementary Figure to be shown, see point n.1) and not in young mice (Suppl. Fig. 3f-h), the expression pattern of the main claudin isoforms, and especially claudin-1, in the different cell populations (see Suppl. Fig. 3b, or by IF analysis) in young vs old and WT vs KO mice must be provided, to provide a mechanistic basis for the observed TJ barrier phenotype. This would help to determine if the phenotype is linked to altered claudin expression or to altered (increased) perijunctional tension.

We agree and will perform immunostainings of Claudins in the young and old KO mice.

3. Mechanistically, the authors mention in the Discussion that Piezo1 might act through RhoA signaling. In Rüksam et al 2017 the authors showed that the uppermost viable layer of the skin has increased apical junctional tension, due to anisotropy of AJ distribution which correlates with EGFR activation and localization. In this context, it is important to know if KO of Piezo-1 affects EGFR localization and signaling, and to probe the RhoA pathway using for example the ROCK inhibitor, instead of blebbistatin.

We will perform immunostainings of EGFR localization and activity and test the involvement of RhoA activity by inhibiting its activity.

Minor comments:

1. The Methods sections should be improved with additional details. For example, the description of quantification of junctional labeling is vague, and there is often no or little indication in the Legends that specifies number of experiments and junctional segments. In addition, quantification of junctional stainings for specific proteins should be done using a junctional reference marker and not as "absolute" values, because there can be variability of staining between samples and experiments. This is especially important when measuring ZO-1, which is a dual AJ-TJ protein (for example at zipper-like junctions ZO-1 colocalizes with AJ markers). Double labelling with a true TJ marker (occludin or cingulin) and/or a true AJ marker (PLEKHA7, afadin, Ecadherin or a catenin) and quantifying junctional labeling by ratio is highly recommended. This is particularly important when evaluating tension-sensitive epitopes/antigens (alpha-catenin, vinculin, etc)

We will edit the methods section for detail. For the alpha-catenin and vinculin stainings we used ZO-1 masks and quantified junctional labeling from these.

2. Please use ZO-1 (and ZO-2) consistently, instead of ZO1 (or ZO2), which is completely inaccurate.

We ensure use of ZO-1 throughout.

3. Please cite Furuse et al 2002 JCB (see above).

We will include this citation.

4. Please include statistical data in Figure Legends, specifying the number of separate experiments and number of samples. At least three experiments is recommended.

This information was already provided in the manuscript but we will ensure that this information is clear. All experiments have been repeated at least three times.

5. At the end of the introduction the authors mention "putative" occludin-containing TJs. I would delete putative. Epithelial junctions that contain a continuous circumferential linear distribution of occludin/ZO-1/cingulin and form a barrier comply with the definition of a TJs (Citi et al JCS 2024) .

We will remove "putative".

6. Please insert page numbers in the manuscript.

We will include page numbers.

Significance

The notion that mechanosensitive calcium channels contribute to the formation of continuous apical junctions (repair and assembly) was introduced by the Miller lab, using Xenopus oocytes. This manuscript provides a significant conceptual advance, not only by using in vitro and in vivo mouse (mammalian) epidermal keratinocytes as model system, but especially by using Piezo1-KO and rescue experiments, which was not done in the Xenopus model.

This research would be of great interest to cell biologists interested in epithelial differentiation, polarization and junction assembly, and to clinicians that are interested in the molecular basis of skin pathophysiology.

My expertise is in the biochemistry, cell biology and mechanobiology of epithelial junctions. I have used Xenopus embryos, cultured epithelial cells, primary keratinocytes and keratinocyte cell lines and KO mice as model systems. The research of my group focuses on how specific cytoskeletal proteins are organized to transmit forces and are recruited to junctions, and how junctional proteins respond to mechanical force. I have experience in all of the methods described in this paper, except for transepidermal water loss measurement, in situ RNA hybridization and mechanical stretching experiments.

Reviewer #3 (Evidence, reproducibility and clarity (Required)):

This manuscript addresses the important topic of cell-cell junction maturation and mechanical stability, with a specific focus on how mechanotransduction through the Piezo1 channel regulates these processes. The authors present compelling in vivo evidence demonstrating that Piezo1 plays a role in junction stability and barrier function, particularly in aged tissue. The work makes a valuable contribution to our understanding of mechanotransduction in epithelial biology. However, several aspects of the mechanistic model and in vitro experiments require additional development to fully support the authors' conclusions.

Major Strengths:

- The in vivo experiments are well-designed and provide convincing evidence for Piezo1's role in barrier function

- The study identifies an important connection between mechanical sensing and junction maturation

- The age-dependent phenotype provides interesting insights into tissue mechanics

We thank the reviewer for this positive assessment of our work and finding the study carefully done and executed. We are also grateful for the expert suggestions that helped us to further strengthen the manuscript.

1. Areas Requiring Additional Development:

a. Mechanistic Model Definition

A major issue is that the central concept of Piezo1 "balancing membrane and cortical tension" requires more precise definition and experimental support. The authors need to clearly explain what this balance means mechanistically and how it is achieved.

We appreciate this feedback. We will clarify in the text that this balancing entails coordinated regulation of cortical tension and membrane tension in space and time, and timely maturation of junctions requires transient decrease in local cortex tension and a corresponding elevation in membrane tension.

We will further perform additional experiments to better demonstrate this relationship: we will measure cortex tension in cells where membrane tension is elevated using osmotic pressure and vice versa, we will measure membrane tension where cortex tension is reduced using myosin

inhibition. Further, we will extend our mechanical measurements to quantify tension across junctions in wild type and Piezo1-deficient cells using laser ablation.

b. Localization-Function Discrepancy

There is an important inconsistency between the authors' claims about Piezo1's role and its localization: while they conclude that Piezo1 is crucial for mechanical stability, they also show that Piezo1 is not localized at mature junctions. This apparent contradiction needs to be addressed with a clear mechanistic explanation.

We realize that we have not been sufficiently clear in describing the results of the mechanical stability experiments. As can be seen in Fig. 5e, mechanical stretch triggers the remodeling of mature junctions into the "high tension" zipper-like state also in the wild type cells. Piezo1 is found to localize into these zipper-like adhesions (Fig. 2g), so we do not see a discrepancy between the localization and the function of Piezo1 to regulate the stability of these zipper-like junctions. We will clarify this in the text.

c. Quantification and Statistical Analysis

Several key conclusions would benefit from more rigorous quantification:

- The quantification of junction maturation in Fig. 1a and 2a should include independent analysis of each experiment rather than pooling cells from multiple experiments

We will replot the data to show independent experiments.

- Actin morphology and pMLC2 levels at junctions in Fig. 1 need systematic quantification

We will provide these quantifications.

- Cytoskeletal dynamics and morphological changes in Piezo1-eKO cells (Fig. 2a) require quantification

We will provide these quantifications.

d. Methodological and Timeline Clarity

The analysis methods and temporal aspects of several experiments need better documentation:
Analysis Methods:

- The quantification method for mature adhesions (used in Figs. 1a, 1e, 1f, 2a) needs clarification. The Methods section states that "The transition from zipper-like adhesions to mature continuous intercellular junctions were quantified manually," but crucial details are missing:

- * What specific criteria defined a "continuous junction"? * Was this based on complete visibility of the cell perimeter as one junction?

- * How were cells classified as having continuous versus zipper-like adhesions?

We classified junctions based on line scans that were drawn in the direction of the junction. When pixel intensity was constant, this junction was classified as continuous. If the line scan showed an oscillatory pattern the junction was classified as zipper-like. We will provide this information in the Methods section.

- e. The protein intensity quantification at junctions requires methodological clarification. The Methods state "For quantifying intensities at junctions, max projection images were generated, and region of interests (ROIs) were restricted to ZO1-positive junctions." However:

- * Were ROIs drawn empirically by the user? Or was the ZO-1 signal used to make a mask?

- * Was there an automated step to determine junctional areas (e.g., intensity threshold)?

- * Was the analysis blinded?

If subjective methods were used, this should be clearly stated and potential variability addressed. The analyses were semi-automated: ZO1 signal was used to make a mask and the mask was hand-corrected when necessary. The analyses were not blinded. We have added this information in the text.

2. Timeline Documentation:

- For blebbistatin experiments (Fig. 1e), specify observation timeframes and quantify the extent of accelerated maturation

- The hypotonic shock experiment (Fig. 3e) timeline needs clarification:

- * When were measurements taken relative to Ca²⁺ switch?

- * Duration of hypotonic media exposure?

* *Were there time-dependent effects in cell response?*

We will add more details and quantification into the description of the experiment. Cells were analyzed 8h post calcium switch corresponding to the transition from zippers to belts that occurs at this time. Hypotonic media was added 30 min before measurements to ensure that the cells had sufficient time to remodel the adhesions. We did not analyze additional time points.

3. Data Support and Interpretation

a. *Several conclusions require additional support or clarification:*

- *The claim about "more dynamic cytoskeletal motion and irregularly shaped" cells (Fig. 2a) is not supported by the provided data. Quantification of dynamics and cell shape are needed to support this conclusion. Cytoskeletal imaging data would also be useful.*

We will add quantification and live imaging of the cytoskeleton.

b. *The interpretation of junctional tension requires revision:*

* *Current conclusions about increased junctional tension are inferred indirectly from vinculin (Fig. 1c) and a18-catenin (Fig. S1a) immunostaining images.*

* *Consider either:*

a) *Adding direct junctional tension measurements (e.g., optical measurements, PMID 31964776)*

b) *Limiting claims to well-supported morphological differences and moving tension-related interpretations to the Discussion as speculative elements*

We will perform laser ablation for more direct measurement of tension across junctions.

c. *The description "Analysis of vinculin translocation to intercellular junctions showed reduced levels of vinculin at cell-cell contacts, but abundant vinculin at cell-matrix adhesions (Supplementary Fig. S2a), indicating abnormal build-up of stresses at intercellular junctions of Piezo1-eKO cells" needs revision:*

* *"Build-up" suggests higher tensions in Piezo1-eKO cells, which contradicts impaired adhesion maturation findings. Suggest replacing with "distribution" or "organization"*

* *"Intercellular" is used ambiguously to include both cell-cell and cell-matrix adhesions*

We agree that these terms can be seen as ambiguous and will adjust the text according to these suggestions for clarity.

4. Literature Context:

- *The discussion should incorporate recent relevant literature on Piezo1's role in tight junction regulation (e.g., PMID 37005489, PMID 33636174, PMID 31409093)*

We will include these references.

5. Technical Considerations

- *For localization studies (Fig. 2), using keratinocytes from Piezo1-tdTomato mouse (JAX #029214) would be preferable to heterologously-expressed Piezo1-FLAG, as it would avoid potential artifacts from non-physiological expression levels*

While we acknowledge that these mice could be useful, obtaining these mice for localization studies is a very time consuming and costly endeavor. JaX does not provide live mice, only cryopreserved stocks, so live mice would be ready for shipment earliest 12 weeks from now 12 weeks for a price of 5000 \$/ individual excluding shipment, after which they would have to be expanded and bred for experiments. First experiments would most likely be possible in 4-6 months. Thus, we see that this suggested experiment is beyond the scope of the current study.

Importantly, we express Piezo1 in the Piezo-null background, so this is not a pure overexpression system. In addition, we do not see variability in location in high and low overexpressing cells. Finally, similar junctional localization has been reported previously by others (<https://www.nature.com/articles/s42003-023-04706-4>).

- *Supp Fig. 1b requires additional replicates*

We will add a third replicate.

- The Fig. 3A legend states "Note increase in FLIPPER-TR lifetime indicative of elevated membrane tension in Piezo1-eKO" when the data actually shows the opposite - a decrease in Flipper-TR lifetime indicating lower membrane tension
We apologize for this typological error that will be corrected.

6. Conceptual and Experimental Clarity Needed

Several statements require clearer explanation or additional supporting evidence:

a. Regarding junction maturation mechanisms:

- The authors state: "This indicated that formation of belt-like adhesions was associated with initial contractility build-up by actomyosin stress fibers linked to junctions, followed by a switch to parallel actomyosin bundles and reduced contractility at adhesions, while the junctions themselves were stabilized in a stressed state indicated by a strengthened actin-junction link." Each part of this claim needs experimental support:

* The "initial contractility build-up by actomyosin stress fibers linked to junctions" needs to be demonstrated

* The "switch to parallel actomyosin bundles and reduced contractility at adhesions" requires quantification

* The claim about "junctions themselves were stabilized in a stressed state" needs stronger evidence

We will perform quantitative live imaging of actin dynamics and laser ablation of junction tension to more conclusively address these effects. In addition, we will add quantifications.

b. The statement "contact expansion from zippers to a belt requires collaborative regulation of adhesion tension and actomyosin cytoskeleton to lower interfacial tension at the contact" is unclear and needs clarification

We apologize for the lack of clarity. According to the current biophysical framework and supported by data, cell contractility and thus cortical tension reduce the length of cell junctions, while adhesion tends to extend the length of the junctions (Lenne et al Dev Cell 2021 PMID 33453154). In this model cortical tension contributes positively to the effective surface tension, while adhesion contributes negatively. We will reformulate to use the term "effective surface tension".

c. The claim "Concomitant with emergence of continuous junctions (8h), the stress fibers were replaced by thick actin bundles positioned perpendicularly to junctions (Fig. 1b)" is not clearly supported by the data

We will perform quantitative live imaging of actin dynamics to more conclusively address these effects.

7. Regarding experimental interpretation:

- In Fig. 1e, the authors claim that 5 μ M blebbistatin accelerates junction maturation, but this conclusion is not supported by the statistics ($p = 0.0784$). Additionally, the timeframe of observation and the quantification of maturation speed should be specified

We would like to respectfully point out that the p-value is only one aspect of quantitative analyses and the biological effect, which in this case is clear and reproducible, is viewed by most statistical experts as more relevant, and that the importance of p-values is overstated by many biologists (see official statement from American Statistical Association

<https://www.nature.com/articles/nature.2016.19503>). We can perform one additional experiment to satisfy the reviewer.

- The results section describing Fig. 3 presents seemingly disconnected observations without clear mechanistic links between them, making it difficult to follow the authors' logic and support their conclusions

- The mechanism by which both reduced contractility (blebbistatin) and increased membrane tension can accelerate maturation (Fig. 1e, f; and also in Piezo1-eKO Fig. 3d, e) needs explanation. The fact that these interventions also accelerate maturation also in Piezo1-eKO suggests a mechanism independent of Piezo1 which is at odds with their broad conclusion that Piezo1 balances membrane tension and cortical contractility in the maturation process. The precise mechanism of Piezo1's role in sensing membrane and cortex tension requires clarification.

We apologize for the lack of clarity. According to the current biophysical framework of adhesion mechanics and supported by data from many labs, cell contractility and thus cortical tension reduce the length of cell junctions, while adhesion tends to extend the length of the junctions

(Lenne et al Dev Cell 2021 PMID 33453154). Our finding that moderately reducing contractility by blebbistatin accelerates formation of mature, linear junctions is consistent with this data. On the other hand, membrane tension has been described to constrain cell protrusion dynamics, with high tension decelerating protrusion and low tension facilitating protrusion (Tsuji et al. Nat Cell Biol 2015 PMID: 25938814; Raucher and Sheetz J Cell Biol 2000 PMID 10629223). Consistently, a robust increase in plasma membrane tension is observed transiently for example during mouse embryonic fibroblast cell spreading on fibronectin-coated substrate. Here, high membrane tension physically constrains the lamellipodium positioning adhesions in the leading edge in a myosin II-independent but vinculin-dependent manner to initiate formation of focal adhesions (Pontes et al., J Cell Biol 2017; PMID 28687667). Thus, it is conceivable that a similar mechanism is at play at adherens junctions where high membrane tension promotes cell-cell adhesion molecule positioning to extend the length of the junctions.

Finally, we interpret the result of these experiments similarly as the reviewer does: experimental elevation of membrane tension or decrease in cortical tension bypasses the need for Piezo1, explaining the rescue effect.

We will extensively edit the manuscript for clarity.

- How Piezo1 maintains mechanical stability of mature junctions despite not being localized there needs to be explained

As discussed in an earlier response the term mature is misleading here as tensile stretch triggers mature junctions to reorganize into zipper-like junctions (where Piezo1 localizes) that require Piezo1 for their stability. We will edit the text for clarity.

8. Suggested Additional Experiments:

a. Optional: Given the age-dependent tissue stiffness effects proposed by the authors, examining keratinocyte behavior in vitro on substrates of varying stiffness would provide valuable insights

b. Optional: Direct measurements of tension at cell-cell junctions where Piezo1 localizes would help validate the proposed mechanical model

a. We have made several attempts to perform the proposed substrate rigidity experiments. Here it is critical to point out that we are using primary keratinocytes isolated directly from the mouse. These cells are very challenging to culture and despite a number of attempts on various hydrogels, we have not been able to achieve confluent cell monolayers that are required for such experiments with these cells.

b. We will perform laser ablation to quantify tension at cell-cell junctions.

9. Minor Points:

- The cell biology sections, particularly descriptions of in vitro experiments, would benefit from a thorough revision to improve precision and clarity. For instance, the Results section describes "Analysis of vinculin translocation to intercellular junctions" when no translocation is actually being studied

- Figure legends should clearly indicate what individual data points represent

- Several conclusions are overstated. For example, the authors conclude that "Piezo1 controls the maturation process" and that "Piezo1 is required for cell junction maturation into junctional belts" based on Fig. 2. These are exaggerated claims since maturation still progresses in Piezo1's absence, just more slowly. "Regulates" or "modulates" would be more appropriate terminology

We will modify the text for more precision and clarity.

In conclusion, while this manuscript presents important findings regarding Piezo1's role in junction maturation and stability, addressing the mechanistic and quantification issues outlined above is essential for supporting the authors' conclusions. The authors have laid groundwork for understanding an important biological process, and addressing these points would help readers better appreciate the significance of their findings.

Reviewer #3 (Significance (Required)):

General Assessment: This study investigates the critical role of mechanosensing in epithelial barrier formation and maintenance, with a particular focus on Piezo1's contribution to junction maturation and stability. The work's primary strengths lie in its compelling in vivo demonstrations of Piezo1's importance for barrier function, particularly in aged tissue, and its identification of a novel connection between mechanical sensing and junction maturation. The age-dependent

phenotype provides valuable insights into tissue mechanics and barrier maintenance. However, the mechanistic understanding of how Piezo1 coordinates these processes requires further development, particularly regarding the proposed balance between membrane and cortical tension.

Advance: This work provides several important advances:

1. First demonstration of Piezo1's role in regulating the maturation of cell-cell junctions from zipper-like to belt-like structures
2. Novel insights into how mechanical forces influence junction maturation through mechanosensitive ion channels
3. Important connection between aging, tissue mechanics, and barrier function
4. Integration of mechanical sensing with junction assembly and stability

The findings extend our understanding of epithelial barrier formation beyond traditional molecular pathways to include mechanotransduction, suggesting new therapeutic possibilities for barrier dysfunction. The age-dependent phenotype is particularly significant as it reveals how mechanical properties of tissue influence barrier maintenance over time.

Audience: This research will be of broad interest to multiple communities:

- Cell biologists studying junction assembly and epithelial organization
- Mechanobiologists interested in force transmission and sensing
- Ion channel researchers interested in the physiological roles of channels
- Aging researchers investigating tissue barrier function
- Bioengineers developing therapeutic strategies for epithelial barriers

The findings have both basic research and translational implications, particularly for understanding and treating age-related barrier dysfunction in epithelia.

Reviewer Expertise: Cell biology, mechanobiology, live cell imaging, quantitative image analysis, ion channels

I have sufficient expertise to evaluate all aspects of the manuscript except for the specific age-related physiological changes in mouse skin, which falls outside my area of expertise.

3. Description of the revisions that have already been incorporated in the transferred manuscript

No revisions have been incorporated yet.

4. Description of analyses that authors prefer not to carry out

Please include a point-by-point response explaining why some of the requested data or additional analyses might not be necessary or cannot be provided within the scope of a revision. This can be due to time or resource limitations or in case of disagreement about the necessity of such additional data given the scope of the study. Please leave empty if not applicable.

We would respectfully suggest that obtaining the Piezo1 reporter mouse line to replace the transient transfections of Piezo1-FLAG to report junctional localization of Piezo1 are beyond the scope of the manuscript. Transfections of tagged proteins to rescue null phenotypes and to analyze protein localizations are a widely used method in cell biology and it is not a standard to expect knocking mouse studies for such experiments.

Original submission

First decision letter

MS ID#: jcs.263938

MS TITLE: Piezo1 balances membrane and cortex tension to stabilize intercellular junctions and maintain epithelial barrier

AUTHORS: Ahsan Javed; Aki Stubb; Clementine Villeneuve; Satu-Marja Myllymäki; Franziska Peters; Matthias Rübsam; Carien M Niessen; Leah C Biggs; Sara A Wickström

ARTICLE TYPE: Review Commons Transfer

Dear Dr Wickström,

Many thanks for transferring your manuscript to Journal of Cell Science from Review Commons. I have now had the chance to review your documents and find your plan to be highly responsive. We would like to invite you to revise your manuscript according to your revision plan. Once we receive the revised version, we may seek further input from the Review Commons referees. If you have any questions about this process, please do get in touch.

First revision

Author response to reviewers' comments

We thank the reviewers for the overall very positive assessment of our work and finding the study carefully executed and representing an important advance in the field. We are also grateful for the expert suggestions that helped us to further strengthen the manuscript. As outlined in detail below, we have now addressed the reviewer comments in full with extensive new experimentation and careful editing of the manuscript. The significant new data include:

1. Direct mechanical measurements (laser ablation, additional membrane and cortex tension measurements, and more detailed quantification of myosin and actin) to characterize the mechanical phenotype of the Piezo1-deficient cells in more detail and to delineate how Piezo1 controls balancing membrane and cortex tension to facilitate the timely formation of mature tight junctions.
2. Additional inhibitor experiments and analyses of EGFR activity elucidate the molecular mechanisms by which Piezo1 regulates cell mechanics.
3. Additional TJ marker stainings and quantifications to more precisely delineate the barrier defect.
4. Additional *in vivo* characterization of the Piezo1 null mouse phenotype to more clearly dissect the inflammatory response.

Reviewer #1

Evidence, reproducibility and clarity

*The studies described in this manuscript investigated the mechanical regulation of tight junction (TJ) maturation in the epidermis using a combination of *in vitro* and *in vivo* analysis. The findings indicate that during calcium-induced cell-cell adhesion in keratinocytes, there is an initial build up cortical tension in the actin cytoskeleton, followed by an increase in membrane tension, which is required for formation of mature TJs. The studies also demonstrate that loss of Piezo1 delays TJ maturation via defects in membrane tension. Loss of Piezo1 also impaired epidermal homeostasis and barrier function in aged mice. The authors propose that the balance in forces between the cortex and membrane is essential for TJ assembly and is mediated by Piezo1.*

Overall, the studies are carefully designed and executed and provide a clear role for membrane tension and Piezo1 in TJ development, making use of molecular forces sensors, imaging, and chemical and genetic perturbations. However, not all of the conclusions are fully supported by the data, and some key findings require additional quantitative and statistical analysis.

We thank the reviewer for this positive assessment of our work and finding the study carefully done and executed. We are also grateful for the expert suggestions that helped us to further strengthen the manuscript.

1. *The statement at the end of page 5 ("This indicated that formation of belt-like...") is somewhat overinterpreted from the data shown. To draw conclusions about a switch to reduced contractility at adhesions requires more careful spatio-temporal quantification of F-actin and myosin beyond the example single cells shown in 1b. It would also help to see the localization of Ecadherin during this process.*

We appreciate this feedback and have now included additional quantitative data of the spatiotemporal regulation of F-actin and myosin activity (pMLC2) during junction maturation and how this is impacted by Piezo1 deletion. This new data (new Figures. 1c and 3d) shows that while control cells re-orient their actin filaments from perpendicular orientation with respect to junctions to a parallel configuration and downregulate their myosin activity when the junctions mature from zipper-like to belt-like junctions, this process is delayed in the Piezo1 deficient cells (new Fig. 3d). E-cadherin localization is not affected, as already shown, and it localizes to zippers in both genotypes.

2. *To avoid confusion, the authors should pay careful attention to terminology and be specific when referring to adherens junctions or TJs, rather than just junctions generally.*

We apologize for the lack of clarity. We have used the term intercellular junctions when we refer to cell-cell adhesions in general. We have carefully edited the text to make clear where we refer to adherens junctions (in conjunction with E-cad staining) and only refer to tight junctions in the context of claudin- or occludin-positive junctions).

3. *The labelling of Figure 2b could be clearer. Were the CNL cells also transfected with Piezo1 or mock transfected to control for general effects of transfection? This was not clear from the figure captions.*

We had also transfected CNL cells as control but did not include this data in the manuscript. We have now included this data in the revised manuscript (new Supplementary Fig. S2e) to demonstrate that the effects are not due to transfection artefacts.

4. *In Figure 2c-g it is not specified which timepoints the images represent, and the qualitative description of changes in localisation require quantification.*

We have included time point information and quantifications (new Fig. 2i).

5. *The importance of Piezo1 in junction maturation is somewhat overstated throughout. While Piezo KO clearly delays TJ maturation, the process can still be completed. In the absence of Piezo1 what triggers the rise in membrane tension? Could there be any compensation from Piezo2?*

We thank the reviewer for raising this point as it appears that we have not been clear enough in describing our conclusions. The point of the reviewer is precisely the conclusion we have intended to convey - Piezo1 regulates the timely maturation of junctions but in the end these junctions are formed. However, the mechanical stretch experiments in Fig. 5e clearly show that if exposed to extrinsic stress, these junctions are not mechanically stable. Further, the KO mice display a clear barrier defect, showing that lack of Piezo1 compromises the function of the junctions, possibly through increased mechanical stress that occurs during aging. We have carefully edited the text to better convey this message and avoid the impression that we claim that Piezo1 is essential for junction formation.

We propose that rather than being a key determinate of absolute membrane tension, Piezo1 is required to adjust cell surface tension (cortical tension and membrane tension) to allow two adjacent cells to generate a continuous, stable interface that allows belt-like, high-tension, mature junctions to form in a timely manner. In this scenario there might not be major compensation from an alternative pathway, the process will still occur, it is simply less efficient. Such lack of compensation is indicated by our findings that the requirement of Piezo1 can be bypassed by elevating membrane tension or by reducing cortex tension We have clarified this conclusion in the discussion section.

Importantly, in the epidermis, Piezo2 is not expressed in the epidermal keratinocytes but its expression is restricted to the specialized sensory organ keratinocytes, the Merkel cells (Woo et al., Nature 2014; PMID 24717433). We further addressed if Piezo2 mRNA becomes upregulated in the Piezo1 KO but did not observe this (Figure 2 for reviewers), so we reasoned that it is unlikely that Piezo2 is compensating, although we have not formally ruled that out. We have added this aspect to the discussion.

Figure 2 for Reviewers.

qPCR analysis of Piezo1 and Piezo2 mRNA expression in CNL and Piezo1-eKO keratinocytes. Piezo2 is not expressed in CNL cells and does not become upregulated upon Piezo1 deletion (n=2 independent experiments)

6. Some of the differences noted are subtle and not strongly significant, such as K6 expression, Ca⁺⁺ induced Piezo1 expression, and F4/80 staining. The conclusions related to these responses should be tempered or qualified.

We have performed new stainings for additional macrophage and T-cell markers to strengthen the conclusions of an inflammatory response (new Supplementary Fig. S5c). We have further carefully edited the manuscript to emphasize that these effects are relatively subtle.

7. Analysis of the immune infiltration and the suggested inflammatory response in aged mice is fairly preliminary and not well supported by the data. A second marker of macrophages and addition of T cell markers would help clarify the type of immune response. It would also help to describe the localisation of specific immune cells in more detail and include a direct marker of inflammation (e.g. inflammatory cytokines).

As mentioned above, we have performed new stainings for additional macrophage and T-cell markers to strengthen these conclusions (new Supplementary Fig. S5c).

8. OPTIONAL: Although not essential for the conclusions of the study, the impact and insight could be improved by providing more analysis of the mechanism for the role of Piezo1. For example, does the build up of cortical tension trigger changes in ion channel signalling, and how does this then regulate membrane tension? Is RhoA or aPKC involved?

We appreciate these suggestions. Due to the differences in time scales - adhesion maturation takes hours vs ion channel signaling as visualized with for example Calcium reporters occurs in seconds, it has been challenging to combine these two readouts to understand the relationship between ion channel activity, cortical tension build-up and adhesion maturation. Calcium imaging requires frame rates in the second time scale, which precludes long term imaging due to phototoxicity. We have analyzed the involvement of RhoA by inhibiting its upstream kinase ROCK1 with Y27632. These new experiments show that inhibition of ROCK1 restores timely formation of mature junctions (new Supplementary Fig. 3a). Further, we have analyzed membrane tension in response to myosin inhibition and observe that inhibition of actomyosin contractility elevates membrane tension in Piezo1-deficient cells to the level of control cells (new Supplementary Fig. 3b). Intriguingly, vice versa, in new experiments where we elevate membrane tension using hypo-osmotic buffer, we observe an attenuation in the enhanced cortex stiffness of Piezo1 deficient cells (new Fig. 3g), highlighting how cortical and membrane tension counterbalance each other and the critical role of Piezo1 in this process.

Significance

The process by which epithelia assemble and maintain effective barriers is complex and requires precise spatio-temporal regulation. This study provides some new insight into the mechanical regulation of TJ assembly within the epidermis. It builds upon previous work that identified essential biomechanical cross-talk between adherens junctions and TJs and adds some new

information on the timings and specific roles of membrane tension and Piezo1. The interplay between cortical and membrane tension is noteworthy, and this mechanism may have important implications in other barrier tissues. A limitation of the study is a lack of mechanistic detail in how the mechanical switch occurs during TJ maturation, including the specific molecules, structures, and interactions with Piezo1.

The study also describes the functional implications, whereby loss of Piezo1 in the mouse disrupts barrier integrity. However, these effects were quite subtle. Barrier homeostasis was only disrupted in aged mice, and *in vitro*, loss of Piezo1 delayed but did not prevent junction maturation. It is therefore interesting to speculate what other mechanisms may be involved in TJ maturation. A potential limitation here is also a lack of detail in the analysis of the inflammatory and immune response in Piezo KO skin.

Reviewer #2

Evidence, reproducibility and clarity

This manuscript describes the role of the mechanosensitive ion channel Piezo1 in epithelial junction assembly, using Piezo-1-KO primary epidermal keratinocytes *in vitro* and mouse skin *in vivo*. The authors conclude that Piezo1 allows balancing of membrane versus cortical tension to stabilize junctions and promote tight junction (TJ) barrier integrity assembly. The conclusion that Piezo1 has an important function in the formation and maintenance of apical junctions of keratinocytes both *in vitro* and *in vivo* is well documented by experiments in WT, KO and rescue cells/tissues where different parameters are carefully measured: protein localization, quantification of mature junctions, membrane tension using the flipper probe, use of the myosin inhibitor blebbistatin, analysis of cortical stiffness by AFM, etc. Although, the physiological relevance and the mechanism through which Piezo operates in young skin are not clear, the authors make reasonable claims, that are not too speculative.

We thank the reviewer for this positive assessment of our work and finding the findings well documented. We are also grateful for the expert suggestions that helped us to further strengthen the manuscript.

Major comments:

1. The Supplementary Figure 4d (panel d) that is described in the Results section is missing. It supposedly shows that 1 year-old Piezo1-eKO mice display an increase in transepidermal water loss, indicating that TJ barrier function is compromised. The Figure legend for the panel is also missing. Please provide the Figure panel and the legend.

The reviewer might have confused the figure labeling, this data is shown in the main Figure 4d.

2. TJ barrier function depends on claudins, and the loss of claudin-1 leads to transepidermal water loss (please cite the relevant paper from the Tsukita lab). Considering that altered TJ barrier function is observed only in 1-yr old mice (Supplementary Figure to be shown, see point n.1) and not in young mice (Suppl. Fig. 3f-h), the expression pattern of the main claudin isoforms, and especially claudin-1, in the different cell populations (see Suppl. Fig. 3b, or by IF analysis) in young vs old and WT vs KO mice must be provided, to provide a mechanistic basis for the observed TJ barrier phenotype. This would help to determine if the phenotype is linked to altered claudin expression or to altered (increased) perijunctional tension.

We agree with this criticism and have performed immunostainings of Claudins in the young and old Piezo1-deficient mice (new Fig. 4e and Supplementary Fig. S4i). These new stainings show mildly reduced claudin-1 intensity at junctions, with some reduction (not statistically significant and highly variable between individual mice) already in young mice and a more robust reduction in aged mice. These experiments confirm that the TJ phenotype progressively worsens from young to aged Piezo1-deficient mice, and provide further molecular explanation for the barrier phenotype.

3. Mechanistically, the authors mention in the Discussion that Piezo1 might act through RhoA signaling. In Rüksam et al 2017 the authors showed that the uppermost viable layer of the skin has increased apical junctional tension, due to anisotropy of AJ distribution which correlates with EGFR activation and localization. In this context, it is important to know if KO of Piezo-1 affects EGFR localization and signaling, and to probe the RhoA pathway using for example the ROCK inhibitor, instead of blebbistatin.

We thank the reviewers for these suggestions. We have now used a ROCK inhibitor (Y27632) and observe that like blebbistatin treatment, inhibiting ROCK rescues the adhesion maturation phenotype of Piezo1-deficient cells (new Supplementary Fig. S3a).

We have also analyzed EGFR activity and consistent with the Rüksam et al 2017 study we observe that while control cells efficiently downregulate EGFR activity upon junction maturation, this downregulation is delayed in Piezo1-deficient cells. However, inhibition of EGFR activity in Piezo1-deficient cells does not robustly rescue junction maturation. Therefore, we conclude that the difference in EGFR activity might reflect the defect in junction maturation kinetics rather than be the primary molecular defect downstream of Piezo (Fig.1 for reviewer). As these experiments remain inconclusive and in our view do not contribute significantly to the manuscript, we would prefer not to include the results in the manuscript.

Fig.1 for reviewer

Analysis of the role of EGFR activity on Piezo1-eKO junction phenotype a, b) Representative western blots (a) and quantification of EGFR phosphorylation in Piezo1-deficient keratinocytes after Ca²⁺ switch. Note delayed downregulation of pEGFR in Piezo1-eKO cells (n=3 independent experiments). c, d) Representative images c) and quantification d) of junction maturation from ZO-1 stained keratinocytes treated with the EGFR inhibitor Gefitinib (n=4 independent experiments).

Minor comments:

1. The Methods sections should be improved with additional details. For example, the description of quantification of junctional labeling is vague, and there is often no or little indication in the Legends that specifies number of experiments and junctional segments. In addition, quantification of junctional stainings for specific proteins should be done using a junctional reference marker and not as "absolute" values, because there can be variability of staining between samples and experiments. This is especially important when measuring ZO-1, which is a dual AJ-TJ protein (for example at zipper-like junctions ZO-1 colocalizes with AJ markers). Double labelling with a true TJ marker (occludin or cingulin) and/or a true AJ marker (PLEKHA7, afadin, Ecadherin or a catenin) and quantifying junctional labeling by ratio is highly recommended. This is particularly important when evaluating tension-sensitive epitopes/antigens (alpha-catenin, vinculin, etc)

We apologize for the lack of details in the methods section have edited the methods section for detail. For the alpha-catenin we used ZO-1 masks and quantified junctional labeling from these. For the vinculin stainings we computed basal to-junction intensity ratios (not absolute values). We have added these details into the Methods section in the manuscript.

2. Please use ZO-1 (and ZO-2) consistently, instead of ZO1 (or ZO2), which is completely inaccurate.

We have ensured use of ZO-1 throughout.

3. Please cite Furuse et al 2002 JCB (see above).

We have included this citation.

4. Please include statistical data in Figure Legends, specifying the number of separate experiments and number of samples. At least three experiments is recommended.

We have ensured that this information is present in all legends. All experiments have been repeated at least three times.

5. At the end of the introduction the authors mention "putative" occludin-containing TJs. I would delete putative. Epithelial junctions that contain a continuous circumferential linear distribution of occludin/ZO-1/cingulin and form a barrier comply with the definition of a TJs (Citi et al JCS 2024).

We have removed "putative".

6. Please insert page numbers in the manuscript.

We have included page numbers.

Significance

The notion that mechanosensitive calcium channels contribute to the formation of continuous apical junctions (repair and assembly) was introduced by the Miller lab, using *Xenopus* oocytes. This manuscript provides a significant conceptual advance, not only by using *in vitro* and *in vivo* mouse (mammalian) epidermal keratinocytes as model system, but especially by using Piezo1-KO and rescue experiments, which was not done in the *Xenopus* model.

This research would be of great interest to cell biologists interested in epithelial differentiation, polarization and junction assembly, and to clinicians that are interested in the molecular basis of skin pathophysiology.

My expertise is in the biochemistry, cell biology and mechanobiology of epithelial junctions. I have used *Xenopus* embryos, cultured epithelial cells, primary keratinocytes and keratinocyte cell lines and KO mice as model systems. The research of my group focuses on how specific cytoskeletal proteins are organized to transmit forces and are recruited to junctions, and how junctional proteins respond to mechanical force. I have experience in all of the methods described in this paper, except for transepidermal water loss measurement, *in situ* RNA hybridization and mechanical stretching experiments.

Reviewer #3 (Evidence, reproducibility and clarity (Required)):

This manuscript addresses the important topic of cell-cell junction maturation and mechanical stability, with a specific focus on how mechanotransduction through the Piezo1 channel regulates these processes. The authors present compelling *in vivo* evidence demonstrating that Piezo1 plays a role in junction stability and barrier function, particularly in aged tissue. The work makes a valuable contribution to our understanding of mechanotransduction in epithelial biology. However, several aspects of the mechanistic model and *in vitro* experiments require additional development to fully support the authors' conclusions.

Major Strengths:

- The *in vivo* experiments are well-designed and provide convincing evidence for Piezo1's role in barrier function
- The study identifies an important connection between mechanical sensing and junction maturation
- The age-dependent phenotype provides interesting insights into tissue mechanics

We thank the reviewer for this positive assessment of our work and finding the study carefully done and executed. We are also grateful for the expert suggestions that helped us to further strengthen the manuscript.

1. Areas Requiring Additional Development:

a. Mechanistic Model Definition

A major issue is that the central concept of Piezo1 "balancing membrane and cortical tension" requires more precise definition and experimental support. The authors need to clearly explain what this balance means mechanistically and how it is achieved.

We appreciate this feedback. We have now clarified in the text that this balancing entails coordinated regulation of cortical tension and membrane tension in space and time, where timely maturation of junctions requires transient decrease in local cortex tension and a corresponding elevation in membrane tension.

We have further performed additional experiments to better demonstrate this relationship: We have measure cortex tension using AFM in Piezo1-deficient cells with and without hypo-osmotic treatment that elevates membrane tension, and observe that the hypo-osmotic treatment restores cortex tension to the level of control cells (new Fig. 3g). Vice versa, lowering cortex tension by inhibition of myosin results in elevation of membrane tension in Piezo1-deficient cells to the level of control cells (new Supplementary Fig. 3b). This demonstrates that Piezo1 is required for the accurate balancing of cortex and membrane tension during junction maturation and that restoring this balance with interfering with either parameter rescues timely junction maturation. We thank the reviewer for helping us to further clarify this mechanism.

b. Localization-Function Discrepancy

There is an important inconsistency between the authors' claims about Piezo1's role and its localization: while they conclude that Piezo1 is crucial for mechanical stability, they also show that Piezo1 is not localized at mature junctions. This apparent contradiction needs to be addressed with a clear mechanistic explanation.

We realize that we have not been sufficiently clear in describing the results of the mechanical stability experiments. As can be seen in Fig. 5e, mechanical stretch triggers the remodeling of mature junctions into the "high tension" zipper-like state also in the wild type cells. Piezo1 is found to localize into these zipper-like adhesions (Fig. 2g), so we do not see a discrepancy between the localization and the function of Piezo1 to regulate the stability of these zipper-like junctions. We have edited the text for clarity.

c. Quantification and Statistical Analysis

Several key conclusions would benefit from more rigorous quantification:

- *The quantitation of junction maturation in Fig. 1a and 2a should include independent analysis of each experiment rather than pooling cells from multiple experiments*

We have replotted the data to show independent experiments (new Fig. 1a and Fig. 2a).

- *Actin morphology and pMLC2 levels at junctions in Fig. 1 need systematic quantification*

- *Cytoskeletal dynamics and morphological changes in Piezo1-eKO cells (Fig. 2a) require quantification*

We appreciate this feedback and have now included additional quantitative data of the spatiotemporal regulation of F-actin and myosin activity (pMLC2) during junction maturation and how this is impacted by Piezo1 deletion. This new data (new Fig. 1c and Fig. 3d) shows that while control cells re-orient their actin filaments from perpendicular orientation with respect to junctions to a parallel configuration and downregulate their myosin activity when the junctions mature from zipper-like to belt-like junctions, this process does is delayed in the Piezo1 deficient cells (new Fig. 3d).

d. Methodological and Timeline Clarity

The analysis methods and temporal aspects of several experiments need better documentation:
Analysis Methods:

- *The quantification method for mature adhesions (used in Figs. 1a, 1e, 1f, 2a) needs clarification. The Methods section states that "The transition from zipper-like adhesions to mature continuous intercellular junctions were quantified manually," but crucial details are missing:*

* *What specific criteria defined a "continuous junction"? * Was this based on complete visibility of the cell perimeter as one junction?*

* *How were cells classified as having continuous versus zipper-like adhesions?*

We apologize for the lack of details here. We manually classified junctions based on line scans that were drawn in the direction of the junction. When pixel intensity was constant, this junction was classified as continuous. If the line scan showed an oscillatory pattern the junction was classified as zipper-like. We have provided this information in the Methods section.

e. The protein intensity quantification at junctions requires methodological clarification. The Methods state "For quantifying intensities at junctions, max projection images were generated, and region of interests (ROIs) were restricted to ZO1-positive junctions." However:

* *Were ROIs drawn empirically by the user? Or was the ZO-1 signal used to make a mask?*

* *Was there an automated step to determine junctional areas (e.g., intensity threshold)?*

* *Was the analysis blinded?*

If subjective methods were used, this should be clearly stated and potential variability addressed.

The analyses were semi-automated: ZO1 signal was used to make a mask and the mask was hand-corrected when necessary. For intensity analyses fully automated methods were used and in the new actin orientation analyses all the cell-cell contacts in the field of view were included minimizing errors or human unconscious biases. As all phenotypes were obvious by eye, blinding was not considered meaningful. We have added details for the quantification as well as a statement regarding non-blinded analyses in the text.

2. Timeline Documentation:

- *For blebbistatin experiments (Fig. 1e), specify observation timeframes and quantify the extent of accelerated maturation*

- *The hypotonic shock experiment (Fig. 3e) timeline needs clarification:*

* *When were measurements taken relative to Ca²⁺ switch?*

* *Duration of hypotonic media exposure?*

* *Were there time-dependent effects in cell response?*

We have added more details and quantification into the description of the experiment. Cells were analyzed 4h, 8h and 12h post calcium switch corresponding to the transition from zippers to belts that occurs at this time as indicated in the figures/figure legends. Blebbistatin or ROCK inhibitor was added for the last 6h and 2h of the experiment, respectively. We have clarified these details in the text and figure legends.

Hypotonic media was added 30 min before measurements to ensure that the cells had sufficient time to remodel the adhesions. We did not analyze additional time points.

3. Data Support and Interpretation

a. Several conclusions require additional support or clarification:

- *The claim about "more dynamic cytoskeletal motion and irregularly shaped" cells (Fig. 2a) is not supported by the provided data. Quantification of dynamics and cell shape are needed to support this conclusion. Cytoskeletal imaging data would also be useful.*

We have now added quantification of the actin changes from the fixed images as well as live imaging of actin cytoskeleton dynamics together with particle image velocimetry analysis to support the conclusion of increased cellular motion (new Fig. 2b and Supplementary Movie 3). Cell shape has been quantified later in the manuscript (Fig. 4b) so to avoid redundancy we have removed statements on shape irregularity when referring to data in Fig. 2.

b. The interpretation of junctional tension requires revision:

* *Current conclusions about increased junctional tension are inferred indirectly from vinculin (Fig. 1c) and α 18-catenin (Fig. S1a) immunostaining images.*

* *Consider either:*

a) Adding direct junctional tension measurements (e.g., optical measurements, PMID 31964776)

b) Limiting claims to well-supported morphological differences and moving tension-related interpretations to the Discussion as speculative elements

We agree and have performed direct measurements of junctional tension using laser ablation at 8h post calcium switch when the junctional morphology difference is most evident. As expected from the vinculin and α 18-catenin stainings, the literature and physical principles of line tension that

indicates linear junctions to be under higher tension than zippers, we observe that Piezo1-deficient cells show lower tension at junctions (new Supplementary Fig. S2b).

c. The description "Analysis of vinculin translocation to intercellular junctions showed reduced levels of vinculin at cell-cell contacts, but abundant vinculin at cell-matrix adhesions (Supplementary Fig. S2a), indicating abnormal build-up of stresses at intercellular junctions of Piezo1-eKO cells" needs revision:

* "Build-up" suggests higher tensions in Piezo1-eKO cells, which contradicts impaired adhesion maturation findings. Suggest replacing with "distribution" or "organization"

* "Intercellular" is used ambiguously to include both cell-cell and cell-matrix adhesions

We agree that these terms can be seen as ambiguous and have adjusted the text according to these suggestions for clarity.

4. Literature Context:

- The discussion should incorporate recent relevant literature on Piezo1's role in tight junction regulation (e.g., PMID 37005489, PMID 33636174, PMID 31409093)

We have included these references.

5. Technical Considerations

- For localization studies (Fig. 2), using keratinocytes from Piezo1-tdTomato mouse (JAX #029214) would be preferable to heterologously-expressed Piezo1-FLAG, as it would avoid potential artifacts from non-physiological expression levels

While we acknowledge that these mice could be useful, obtaining these mice for localization studies is a very time consuming and costly endeavor. JaX does not provide live mice, only cryopreserved stocks, so live mice would be ready for shipment earliest 12 weeks from now 12 weeks for a price of 5000 \$/ individual excluding shipment, after which they would have to be expanded and bred for experiments. First experiments would most likely be possible in 4-6 months. Thus, we see that this suggested experiment is beyond the scope of the current study.

Importantly, we express Piezo1 in the Piezo-null background, so this is not a pure overexpression system. In addition, we do not see variability in location in high and low overexpressing cells. Finally, similar junctional localization has been reported previously by others (<https://www.nature.com/articles/s42003-023-04706-4>).

- Supp Fig. 1b requires additional replicates

We have added a third replicate (new Supplementary Fig. S1b).

- The Fig. 3A legend states "Note increase in FLIPPER-TR lifetime indicative of elevated membrane tension in Piezo1-eKO" when the data actually shows the opposite - a decrease in Flipper-TR lifetime indicating lower membrane tension

We apologize for this typographical error that has been corrected.

6. Conceptual and Experimental Clarity Needed

Several statements require clearer explanation or additional supporting evidence:

a. Regarding junction maturation mechanisms:

- The authors state: "This indicated that formation of belt-like adhesions was associated with initial contractility build-up by actomyosin stress fibers linked to junctions, followed by a switch to parallel actomyosin bundles and reduced contractility at adhesions, while the junctions themselves were stabilized in a stressed state indicated by a strengthened actin-junction link." Each part of this claim needs experimental support:

* The "initial contractility build-up by actomyosin stress fibers linked to junctions" needs to be demonstrated

* The "switch to parallel actomyosin bundles and reduced contractility at adhesions" requires quantification

* The claim about "junctions themselves were stabilized in a stressed state" needs stronger evidence

We have modified the sentence read "while the junctions themselves were stabilized in a tensed state indicated by vinculin recruitment as a proxy for a strengthened actin-junction link." Also, as described in responses to previous points, we have now performed

- quantitative analysis of actin and myosin dynamics (new Fig. 1c).
- laser ablation to directly quantify tension (new Supplementary Fig. S2b).

b. The statement "contact expansion from zippers to a belt requires collaborative regulation of adhesion tension and actomyosin cytoskeleton to lower interfacial tension at the contact" is unclear and needs clarification

We apologize for the lack of clarity. According to the current biophysical framework and supported by data, cell contractility and thus cortical tension reduce the length of cell junctions, while adhesion tends to extend the length of the junctions (Lenne et al Dev Cell 2021 PMID 33453154). In this model cortical tension contributes positively to the effective surface tension, while adhesion contributes negatively. We will reformulate this statement and use the term "effective surface tension".

c. The claim "Concomitant with emergence of continuous junctions (8h), the stress fibers were replaced by thick actin bundles positioned perpendicularly to junctions (Fig. 1b)" is not clearly supported by the data

As described in responses to previous points, we have performed quantitative analysis to better characterize this reorientation (new Fig. 1c).

7. Regarding experimental interpretation:

- In Fig. 1e, the authors claim that 5 μ M blebbistatin accelerates junction maturation, but this conclusion is not supported by the statistics ($p = 0.0784$). Additionally, the timeframe of observation and the quantification of maturation speed should be specified

We would like to respectfully point out that the p-value is only one aspect of quantitative analyses and the biological effect, which in this case is clear and reproducible, is viewed by most statistical experts as more relevant, and that the importance of p-values is overstated by many biologists (see official statement from American Statistical Association

<https://www.nature.com/articles/nature.2016.19503>). We have performed one additional experiment to obtain statistical significance (new Fig. 1f). We have also clarified the time frame of this observation.

- The results section describing Fig. 3 presents seemingly disconnected observations without clear mechanistic links between them, making it difficult to follow the authors' logic and support their conclusions

- The mechanism by which both reduced contractility (blebbistatin) and increased membrane tension can accelerate maturation (Fig. 1e, f; and also in Piezo1-eKO Fig. 3d, e) needs explanation. The fact that these interventions also accelerate maturation also in Piezo1-eKO suggests a mechanism independent of Piezo1 which is at odds with their broad conclusion that Piezo1 balances membrane tension and cortical contractility in the maturation process. The precise mechanism of Piezo1's role in sensing membrane and cortex tension requires clarification.

We apologize for the lack of clarity. According to the current biophysical framework of adhesion mechanics and supported by data from many labs, cell contractility and thus cortical tension reduce the length of cell junctions, while adhesion tends to extend the length of the junctions (Lenne et al Dev Cell 2021 PMID 33453154). Our finding that moderately reducing contractility by blebbistatin accelerates formation of mature, linear junctions is consistent with this data. On the other hand, membrane tension has been described to constrain cell protrusion dynamics, with high tension decelerating protrusion and low tension facilitating protrusion (Tsujita et al. Nat Cell Biol 2015 PMID: 25938814; Raucher and Sheetz J Cell Biol 2000 PMID 10629223). Consistently, a robust increase in plasma membrane tension is observed transiently for example during mouse embryonic fibroblast cell spreading on a fibronectin-coated substrate. Here, high membrane tension physically constrains the lamellipodium positioning adhesions in the leading edge in a myosin II-independent but vinculin-dependent manner to initiate formation of focal adhesions (Pontes et al., J Cell Biol 2017; PMID 28687667). Thus, it is conceivable that a similar mechanism is at play at adherens junctions where high membrane tension promotes cell-cell adhesion molecule positioning to extend the length of the junctions.

To experimentally demonstrate this effect, we have analyzed membrane tension in response to myosin and observe that inhibition of actomyosin contractility elevates membrane tension in Piezo1-deficient cells to the level of control cells (new Supplementary Fig. 3b), indicating that increased cortex tension facilitates low membrane tension. Importantly, vice versa, in new

experiments where we elevate membrane tension using hypo-osmotic buffer, we observe an attenuation in the enhanced cortex stiffness of Piezo1 deficient mice (new Fig. 3g), highlighting how cortical and membrane tension counterbalance each other and the critical role of Piezo1 in this process.

Finally, we interpret the result of these experiments similarly as the reviewer does: experimental elevation of membrane tension or decrease in cortical tension bypasses the need for Piezo1, explaining the rescue effect.

- How Piezo1 maintains mechanical stability of mature junctions despite not being localized there needs to be explained

As discussed in an earlier response the term mature is misleading here as tensile stretch triggers mature junctions to reorganize into zipper-like junctions (where Piezo1 localizes) that require Piezo1 for their stability. We have edited the text for clarity.

8. Suggested Additional Experiments:

a. *Optional: Given the age-dependent tissue stiffness effects proposed by the authors, examining keratinocyte behavior in vitro on substrates of varying stiffness would provide valuable insights*

b. *Optional: Direct measurements of tension at cell-cell junctions where Piezo1 localizes would help validate the proposed mechanical model*

a. We have made several attempts to perform the proposed substrate rigidity experiments. Here it is critical to point out that we are using primary keratinocytes isolated directly from the mouse. These cells are very challenging to culture and despite a number of attempts on various hydrogels, we have not been able to achieve confluent cell monolayers that are required for such experiments with these cells.

b. We have performed laser ablation to quantify tension at cell-cell junctions (new Supplementary Fig. S2b). As predicted by the findings on reduced vinculin and alpha18-catenin at Piezo1 junctions as well as literature pointing to belt-like junction being under high tension, we observe that Piezo1-deficient junctions are under lower tension 8h post calcium switch.

9. Minor Points:

- The cell biology sections, particularly descriptions of in vitro experiments, would benefit from a thorough revision to improve precision and clarity. For instance, the Results section describes "Analysis of vinculin translocation to intercellular junctions" when no translocation is actually being studied

- Figure legends should clearly indicate what individual data points represent

- Several conclusions are overstated. For example, the authors conclude that "Piezo1 controls the maturation process" and that "Piezo1 is required for cell junction maturation into junctional belts" based on Fig. 2. These are exaggerated claims since maturation still progresses in Piezo1's absence, just more slowly. "Regulates" or "modulates" would be more appropriate terminology

We appreciate this feedback and have modified these statements according to the reviewer suggestions.

In conclusion, while this manuscript presents important findings regarding Piezo1's role in junction maturation and stability, addressing the mechanistic and quantification issues outlined above is essential for supporting the authors' conclusions. The authors have laid groundwork for understanding an important biological process, and addressing these points would help readers better appreciate the significance of their findings.

Reviewer #3 (Significance (Required)):

General Assessment: This study investigates the critical role of mechanosensing in epithelial barrier formation and maintenance, with a particular focus on Piezo1's contribution to junction maturation and stability. The work's primary strengths lie in its compelling in vivo demonstrations of Piezo1's importance for barrier function, particularly in aged tissue, and its identification of a novel connection between mechanical sensing and junction maturation. The age-dependent phenotype provides valuable insights into tissue mechanics and barrier maintenance. However, the mechanistic understanding of how Piezo1 coordinates these processes requires further development, particularly regarding the proposed balance between membrane and cortical tension.

Advance: This work provides several important advances:

- 1. First demonstration of Piezo1's role in regulating the maturation of cell-cell junctions from zipper-like to belt-like structures*
- 2. Novel insights into how mechanical forces influence junction maturation through mechanosensitive ion channels*
- 3. Important connection between aging, tissue mechanics, and barrier function*
- 4. Integration of mechanical sensing with junction assembly and stability*

The findings extend our understanding of epithelial barrier formation beyond traditional molecular pathways to include mechanotransduction, suggesting new therapeutic possibilities for barrier dysfunction. The age-dependent phenotype is particularly significant as it reveals how mechanical properties of tissue influence barrier maintenance over time.

Audience: This research will be of broad interest to multiple communities:

- Cell biologists studying junction assembly and epithelial organization*
- Mechanobiologists interested in force transmission and sensing*
- Ion channel researchers interested in the physiological roles of channels*
- Aging researchers investigating tissue barrier function*
- Bioengineers developing therapeutic strategies for epithelial barriers*

The findings have both basic research and translational implications, particularly for understanding and treating age-related barrier dysfunction in epithelia.

Reviewer Expertise: Cell biology, mechanobiology, live cell imaging, quantitative image analysis, ion channels

I have sufficient expertise to evaluate all aspects of the manuscript except for the specific age-related physiological changes in mouse skin, which falls outside my area of expertise.

Second decision letter

MS ID#: jcs263938R1

MS TITLE: Piezo1 balances membrane and cortex tension to stabilize intercellular junctions and maintain epithelial barrier

AUTHORS: Ahsan Javed; Aki Stubb; Clementine Villeneuve; Satu-Marja Myllymäki; Franziska Peters; Matthias Rübsam; Carien M Niessen; Leah C Biggs; Sara A Wickström

ARTICLE TYPE: Review Commons Transfer

Dear Dr Wickström,

Thank you for sending your manuscript to Journal of Cell Science through Review Commons.

I am happy to tell you that your manuscript has been accepted for publication in Journal of Cell Science, pending standard publication integrity checks.